# Population analyses reveal heterogenous encoding in the medial prefrontal cortex during naturalistic foraging

**Ji Hoon Jeong, June-Seek Choi***

School of Psychology, Korea University, Seoul, Republic of Korea

## eLife Assessment

This **important** study by Jeong and Choi studied neural activity in the medial prefrontal cortex (mPFC) while rats performed a foraging paradigm in which rats forage for rewards in the absence or presence of a threatening object (Lobsterbot). The authors present interesting observations suggesting that the mPFC population activity switches between distinct functional modes conveying distinct task variables- such as the distance to the reward location and types of threat-avoidance behaviors-depending on the location of the animal. The reviewers thought that the results are overall **convincing**, appreciated the value of studying neural coding in naturalistic settings, and felt that this work offers significant insights into how the mPFC operates during foraging behavior involving reward-threat conflict.

*For correspondence:
j-schoi@korea.ac.kr

**Competing interest:** The authors declare that no competing interests exist.

**Abstract** Foraging in the wild requires coordinated switching of critical functions, including goal-oriented navigation and context-appropriate action selection. Nevertheless, few studies have examined how different functions are represented in the brain during naturalistic foraging. To address this question, we recorded multiple single-unit activities from the medial prefrontal cortex (mPFC) of rats seeking a sucrose reward in the presence of an unpredictable attack posed by a robotic predator (Lobsterbot). Simultaneously recorded ensemble activities from neurons were analyzed in reference to various behavioral indices as the animal moved freely across the foraging area (F) between the nest (N) and the goal (E) area. An artificial neural network, trained with simultaneously recorded neural activity, estimated the rat's current distance from the Lobsterbot. The accuracy of distance estimation was the highest in the middle F-zone in which the dominant behavior was active navigation. The spatial encoding persisted in the N-zone when non-navigational behaviors such as grooming, rearing, and sniffing were excluded. In contrast, the accuracy decreased as the animal approached the E-zone, when the activity of the same neuronal ensembles was more correlated with events related to dynamic decision-making between food procurement and Lobsterbot evasion. A population-wide analysis confirmed highly heterogeneous encoding by the region. To further assess the decision-related activity in the E-zone, a naive Bayesian classifier was trained to predict the success and failure of avoidance behavior. The classifier predicted the avoidance outcome as much as 6 s before the head withdrawal. In addition, two sub-populations of recorded units with distinct temporal dynamics contributed differently to the prediction. These findings suggest that an overlapping population of mPFC neurons may switch between two heterogeneous modes, encoding relevant locations for goal-directed navigation or an imminent situational challenge.

## Introduction

Foraging in nature entails context-appropriate action selection to resolve a mixture of challenges, including risk and value assessment, maintenance of goal-oriented navigation, and task-switching (*Pyke et al., 1977*; *Stephens and Krebs, 1986*). For example, foraging animals must update the value of the goal in a space-time domain, negotiate threats while obtaining food, and regulate various stimulus-driven defensive behaviors (*Fanselow and Lester, 1988*; *Kim et al., 2018*; *Mobbs et al., 2007*; *Mobbs et al., 2020*; *Mobbs and Kim, 2015*). These survival-related demands require executive functions capable of flexibly updating risk and value representations in working memory to adapt to the foraging environment. Numerous studies have identified the prefrontal cortex as a primary computational hub for hosting multiple strategies and integrating survival-related information (*Fuster, 1973*; *Korn and Bach, 2019*; *Lee et al., 2014*).

A critical component of successful foraging is goal-directed navigation. Although the hippocampus (HIPP) is crucial for maintaining reward location and path planning (*Jarzebowski et al., 2022*; *Krishnan et al., 2022*; *Nyberg et al., 2022*; *Wikenheiser et al., 2013*; *Wikenheiser and Redish, 2015*), the medial prefrontal cortex (mPFC) and its interaction with the HIPP also play a role in goal-directed navigation (*Guise and Shapiro, 2017*; *Sapiurka et al., 2016*; *Spellman et al., 2015*; *Yu et al., 2018*). Furthermore, given that the mPFC is involved in processing deliberate defensive strategies in multiple situations (see *Mobbs et al., 2020* for review), naturalistic foraging where risk and reward coexist may serve as a suitable foundation for exploring adaptive information processing in the mPFC. Notably, mPFC houses spatially selective neurons that represent relative positions in a maze (*Kaefer et al., 2020*; *Yang and Mailman, 2018*), goal locations (*Hok et al., 2005*; but see *Böhm and Lee, 2020*), and precise Cartesian coordinates during food procurement (*Mashhoori et al., 2018*).

The mPFC also processes emotional stimuli, especially those signaling environmental threats. Through its extensive connections with the major limbic areas including the amygdala (*Gabbott et al., 2005*; *Hoover and Vertes, 2007*; *Vertes, 2004*), the mPFC is causally linked to both learning and expression of fear responses. For example, the infralimbic cortex (IL) of the mPFC is associated with the extinction of conditioned fear response (*Park and Choi, 2010*; *Sierra-Mercado et al., 2006*). In contrast, the activation of the prelimbic cortex (PL) is linked to the expression of fear-conditioned responses (*Burgos-Robles et al., 2009*; *Dejean et al., 2016*; *Diehl et al., 2018*; *Kim et al., 2013*). In addition, PL neurons show increased activity in response to survival-related stimuli such as food, artificial predators, or both (*Kim et al., 2018*). These findings suggest that the critical information for risky decision-making might be represented and processed in the mPFC.

Currently, there is limited understanding of how the brain organizes navigation and behavior selections during naturalistic foraging. It is unclear whether spatial representation and emotional processing are separated at the single neuron level or coexist within overlapping populations of the mPFC neurons. In this study, we devised a naturalistic foraging task which allowed self-initiated shuttling between different locations and required avoiding an unpredictable attack by a robotic predator while approaching sucrose. This approach allowed continuous monitoring of the neural activity as the rat switched between navigation and strategic reward procurement. Machine learning-based populational decoding methods, alongside single-cell analyses, were employed to investigate the correlations between neuronal activity and a range of behavioral indices across different sections within the foraging arena.

## Results

### Mix of avoidance and escape behaviors during naturalistic foraging under threat

The foraging arena, modified from *Kimm and Choi, 2018*, was designed to encourage naturalistic foraging that involved both approaching food and avoiding threat. The arena was divided into two distinctive zones by a wall with a pass-through opening in the middle (*Figure 1A*): the foraging (F) and the nest (N) zones. At one end of the F-zone, a motorized gate controlled the access to the robot compartment, which contained a sucrose port and the predator-like 'Lobsterbot', named for its large, claw-like arms. Entry into the sucrose port was detected by a beam sensor mounted near the gate (*Figure 2B*). A semicircular area surrounding the robot compartment was designated as the encounter (E) zone where the rat's body was located during sucrose licking and Lobsterbot attack.

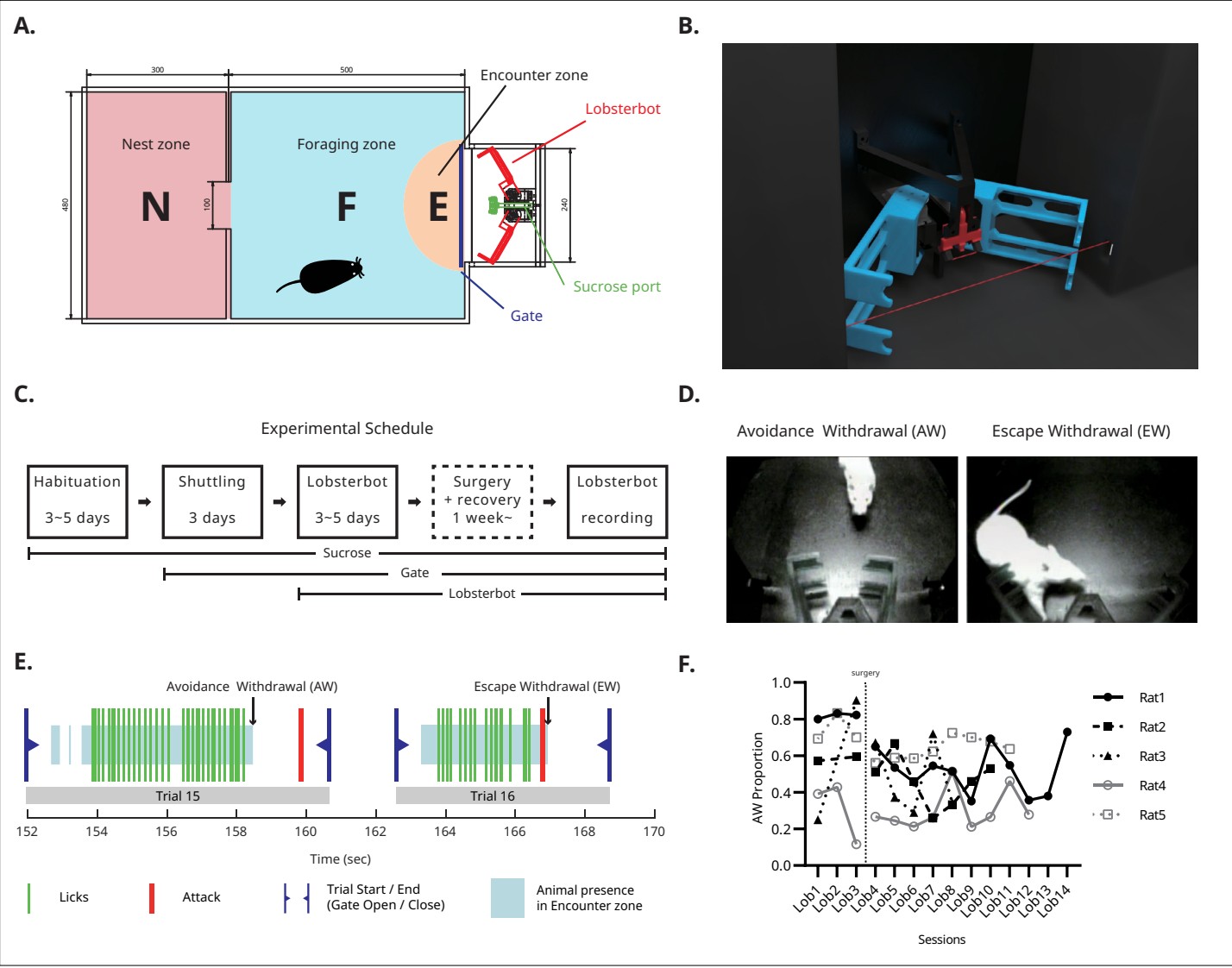

**Figure 1.** Behavioral procedure and apparatus. (**A**) Schematic of the arena showing major sections: N, nest zone; F, foraging zone; E, encounter zone. At the end of the E-zone, the Lobsterbot (red) guards the sucrose delivery port (green). (**B**) Rendered 3-D image of the Lobsterbot. The sucrose port is positioned between the "claws". Two red lines indicate infrared detectors: a short line for lick detection (short line) and a long line for E-zone entry detection. (**C**) Experimental schedule. (**D**) Example snapshots of avoidance withdrawal (AW) and escape withdrawal (EW). In an AW trial, the rat typically retracts its head in advance and observes the Lobsterbot attack. In an EW trial, the rat reflexively flees from the attack. (**E**) Example behavior data containing two consecutive trials (Trials 15 and 16). Each trial started with a reentry to the N zone which triggers gate opening. The rat leaving the N zone typically moves toward the E zone across the F zone. The entry to the E zone is detected by an IR beam sensor (blue shade). Within the E zone, the rat starts licking (green lines) until being attacked by Lobsterbot (red line) 3 or 6 s after the first lick. The rat shows voluntary withdrawal behavior (AW; Trial 15) or forced escape behavior (EW; Trial 16). (**F**) Summary of the AW trial rates for each animal during Losterbot sessions. Points for Lob2 of Rat2 and Rat3 are omitted because they did not approach the robot during the entire Lob2 session.

The online version of this article includes the following figure supplement(s) for figure 1:

**Figure supplement 1.** Head withdrawal time distribution across all subjects, categorized by trial type.

**Figure supplement 2.** Foraging-related behavioral indices fluctuate upon the initial encounter with the Lobsterbot but stabilize after three sessions.

The experiment comprises three distinct phases (*Figure 1C*): habituation, shuttling, and Lobsterbot phase. The Lobsterbot phase was further divided into early and late phases separated by the time of surgery. During the habituation phase, a food-deprived rat was acclimated to the sucrose port that released sucrose solution. The lick port was activated by an IR-beam sensor, triggering the solenoid valve when the beam was interrupted. The rat gradually learned to obtain rewards by continuously

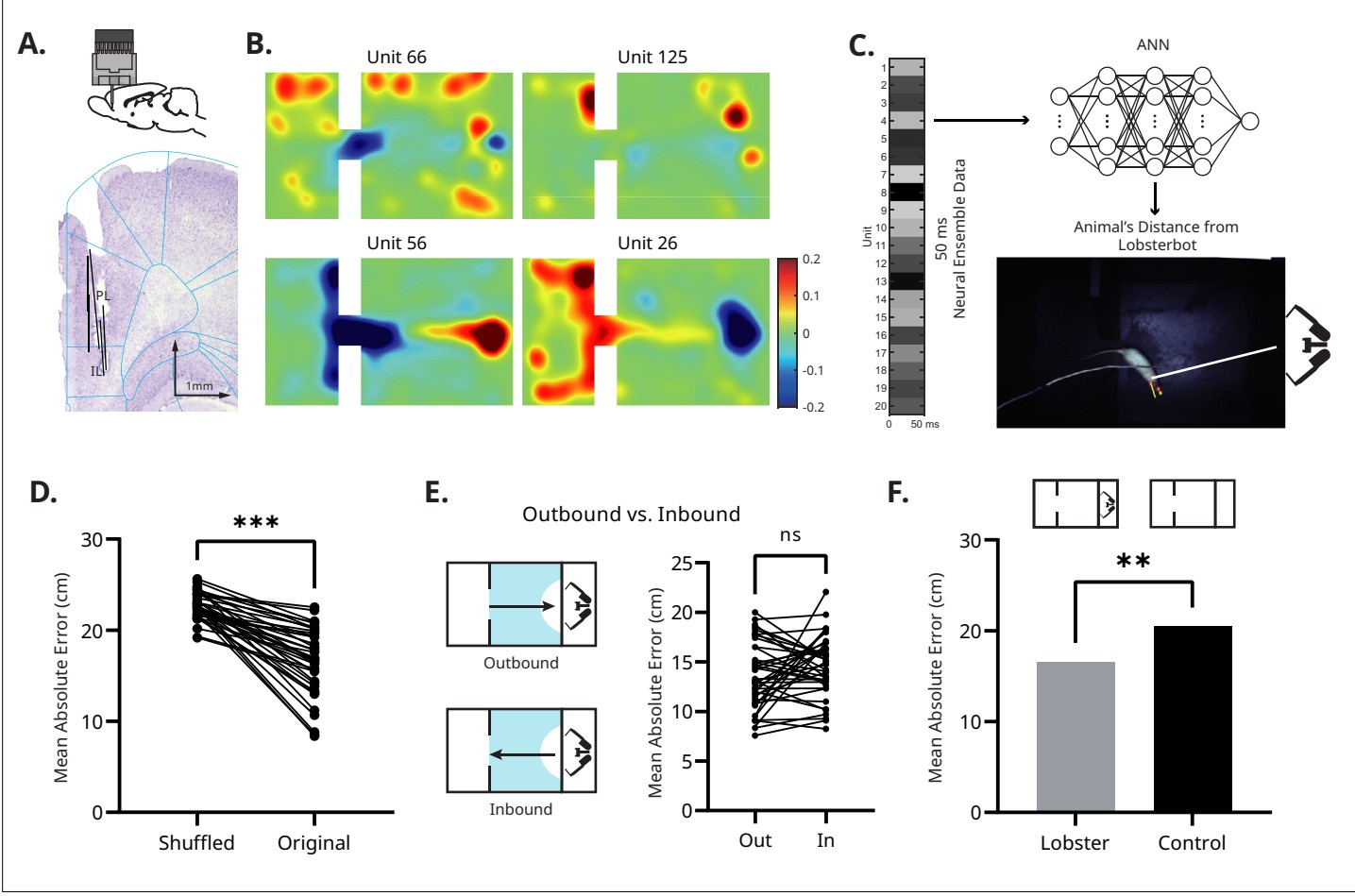

**Figure 2.** Ensemble activity in the mPFC predicts distance from the goal. (**A**) Electrode implantation and recording sites. Top: A movable 4-tetrode microdrive was initially implanted in the PL region and lowered ventrally toward the IL after every recording session. Bottom: Representative recording tracks from all five animals are superimposed over an image of a stained coronal section of the frontal brain. Histological examination of all brain sections confirmed that the electrode tracks spanned the dorsoventral axis between the PL and IL. (**B**) Modulation of unit firing showing place-cell like activities. Units 66 and 125 exhibit fragmented place fields all over the arena, while Units 56 and 26 display relatively large place fields surrounding particular spots such as the gates. Heat maps are calculated from z-scored spatial tuning curves. (**C**) Schematics of the ensemble decoding analysis. The four-layer deep artificial neural network (ANN) receives populational neural data during a 50 ms time window and is trained to predict the rat's current distance from the center of the E-zone. The example data depicted in the figure is a sample recording from 20 units when the rat is at a particular distance away from the center of the E-zone, indicated by the white bold line. (**D**) Accuracy of the distance regressor. Mean Absolute Error (MAE) was significantly smaller for the original dataset (16.61 cm, Original) compared to the shuffled dataset (Shuffled), and below the rat's body length, suggesting that the mPFC ensembles encode spatial correlates of the distance from the goal. Statistical significance was determined using paired *t*-test (***p<0.001). N = 40. (**E**) Prediction accuracy in the F-zone during outbound/inbound paths. Decoding accuracy in the F-zone was calculated separately for the outbound (from the N-zone to the E-zone) and inbound (from the E-zone to the N-zone) paths. The decoding accuracy remained unchanged despite the differences in motivation and perceived visual cues due to the movement direction. Statistical significance was determined using paired *t*-test. N = 40. (**F**) Comparison of the regressor's accuracy from the control experiment. When the Lobsterbot was removed from the robot compartment, reverting the task back to simple shuttling, the mPFC distance regressor's performance significantly decreased compared to the Lobsterbot phase. Statistical significance was determined using unpaired *t*-test (**p<0.01). N = 40 for Lobster; N = 9 for Control.

The online version of this article includes the following figure supplement(s) for figure 2:

**Figure supplement 1.** Comparison between distance regressor algorithms.

licking the port. Once the rat maintained a stable licking frequency, typically after 3 days, the shuttling phase began.

In the shuttling phase, rats were trained to shuttle between zones to obtain sucrose. The robot compartment's gate closed 6 s after the first lick and reopened only when the rat returned to the N-zone. After three days of shuttling phase, the Lobsterbot phase was introduced. During this phase, the robot executed a fast-striking attack, 3 s (30%) or 6 s (70%) from the first lick, with the timing

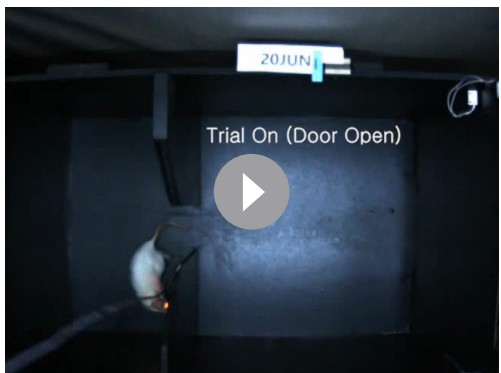

**Video 1.** Single trial structure. Demonstration of the event sequence within a single trial, showing trial onset (door open), head entry, attack, head withdrawal (escape withdrawal), and trial end (door closed).

https://elifesciences.org/articles/93994/figures#video1

randomized across trials to mimic natural uncertainty and prevent simple time-counting strategies. As in the shuttling phase, the gate closed automatically after the attack, and the rat must return to the N-zone to reopen it.

In each trial of the Lobsterbot phase, the foraging rat could respond with either *avoidance withdrawal* (AW: head retraction before the attack) or *escape withdrawal* (EW: reflexive head retraction after the attack; *Figure 1D*). *Figure 1E* illustrates a representative sequence of responses and events in two consecutive trials. AW was marked by a distinct interval between head withdrawal (end of the blue shade) and the attack (red line), whereas EW was characterized by overlap between the attack and the animal's presence in the E-zone (*Videos 1 and 2*).

We examined head withdrawal time points to assess whether rats developed a temporal strategy to differentiate between the 3 s and 6 s attack trials. No such strategy was observed; premature withdrawals during 6 s trials were evenly distributed (see *Figure 1—figure supplement 1*).

All five rats exhibited distinctive phasic behavioral changes. Notably, approach attempts (number of trials) dropped sharply during the first Lobsterbot session (Lob1), then stabilized in later sessions after the surgery (starting from Lob4; *Figure 1—figure supplement 2A*). Similar phasic patterns were observed in the number of licks (*Figure 1—figure supplement 2B*), licks per trial (*Figure 1—figure supplement 2C*), and lick latency after gate opening (*Figure 1—figure supplement 2D*). Comparing the last shuttling session (Sht3) with all pre-surgery Lobsterbot sessions (Lob1-Lob3) revealed significant changes in approach attempts (Friedman test; $\chi^2(3)=9.240$, p=0.017), licks ($\chi^2(3)=12.60$, p<0.001), lick latency ($\chi^2(3)=7.800$, p=0.044), and licks per trial ($\chi^2(3)=13.08$, p<0.001). Specifically, both licks and licks per trial decreased during Lob1 (Dunn's multiple comparisons test; p=0.021; p=0.043) and returned to shuttling-phase levels by Lob3 (p>0.999 for both).

In the early sessions of the Lobsterbot phase, the rats displayed passive defensive behaviors such as freezing (*Blanchard and Blanchard, 1969*; *McClelland and Colman, 1967*) and stretched-attend posture (*Grewal et al., 1997*; *Grant and Mackintosh, 1963*). These behaviors diminished toward the middle of the phase. Compared to the shuttling phase—where rats typically licked until the gate closed on nearly all trials (88.96%), the Lobsterbot phase featured a mix of AW and EW trials (65.11% and 34.89%, respectively). Due to the transient drop in approach attempts and the variety of defensive responses following initial threat exposure, surgery was performed after three Lobsterbot sessions (*Figure 1F*).

## Population activity in the mPFC predicts the distance from the Lobsterbot while navigating in the arena

Recording data were collected from a total of five rats implanted with a custom-made moveable tetrode drive. The tetrodes were initially implanted in the PL and advanced ventrally toward the IL at the end of each session (0.1~0.2 mm/session). Recording locations were reversely calculated from the marking lesion made after the last recording session. Tetrode tracks were identified by visual inspection under a microscope as indicated by black vertical lines on the matching

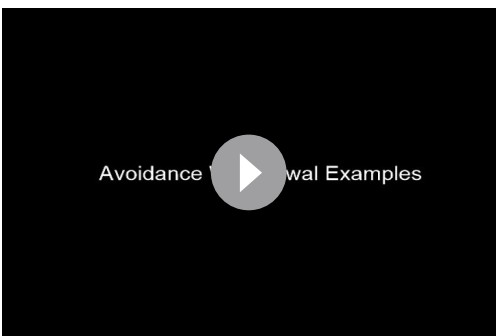

**Video 2.** Examples of avoidance and escape withdrawal. The video shows three examples of avoidance withdrawal trials, in which the animal leaves the E-zone before the attack, and three examples of escape withdrawal trials, in which the animal leaves the E-zone after the attack.

https://elifesciences.org/articles/93994/figures#video2

atlas (*Paxinos and Watson, 2007*) as shown in *Figure 2A*. All tetrodes except for one spanned both the PL and the IL. After cluster cutting and preprocessing (see Materials and methods), 632 units (57~198 units per rat) from the PL (n=379) and the IL (n=253) were selected for ensemble analysis.

Previous studies have reported spatial correlates were observed in multiple subregions of the mPFC (*Hok et al., 2005*; *Jung et al., 1998*; *Ma et al., 2023*; *Mashhoori et al., 2018*; *Sauer et al., 2022*; *Spellman et al., 2015*). To determine whether location-specific neural activity exists at the single-cell level in our dataset, a traditional place map was constructed for individual neurons. Although most neurons did not show cohesive place fields, a few neurons modulated their firing rates based on the rat's current location. However, even these neurons were not comparable to place cells in the hippocampus (*O'Keefe and Dostrovsky, 1971*) or grid cells in the entorhinal cortex (*Hafting et al., 2005*) as the place fields were patchy and irregular in some cases (*Figure 2B*; Units 66 and 125) or too broad, spanning the entire zone rather than a discrete location within it (Units 26 and 56). The latter type of neuron has been identified in other studies (e.g. *Kaefer et al., 2020*).

Since we did not find neural activity that can be mapped straight onto the spatial layout of the arena based on individual activity, we performed machine-learning (ML) analysis targeting all recorded units during the given session. Among various machine learning algorithms, we selected a robust tool for decoding underlying patterns in the data, rather than to model the architecture of the mPFC. We implemented a four-layer artificial neural network regressor (ANN; see Materials and methods for a detailed structure), as the ANN achieves significantly lower decoding errors (*Figure 2—figure supplement 1*) and has robustness to hyperparameter changes (*Glaser et al., 2020*). The inputs to the regressor were the simultaneously recorded neural activity, which was segmented into 50 ms bins, and the predicted output was the distance between the rat's head and the center of the E-zone where reward and threat co-exist (*Figure 2C*).

The analysis of trained ANNs revealed that the distance from the rat to the center of the E-zone can be decoded from the mPFC ensemble activity. Compared to the control regressor trained on a shuffled dataset, the regressor with original data showed significantly lower errors (paired t-test, $t(39)$ = 10.47, p<0.001). However, we did not observe a difference in decoding accuracy between the PL and IL ensembles, and there were no significant interactions between regressor type (shuffled vs. original) and regions (mixed-effects model; regions: p=0.996; interaction: p=0.782). These results indicate that the population activity in both the PL and IL contains spatial information (*Figure 2D*, *Video 3*). Overall accuracy measured by mean absolute error (MAE) was 16.61 (±3.67) cm, which was slightly higher than values from previous reports employing different spatial tasks (*Ma et al., 2023*; *Mashhoori et al., 2018*). This reduced accuracy may reflect the smaller number of neurons per ensemble in our dataset (mean = 15.8) relative to prior studies. Given the properties of the algorithm, we expect that larger ensembles would improve performance.

We next asked whether the decoding accuracy varied with movement directions. Specifically, we compared the decoding accuracy of the regressor for outbound (from the N- to E-zone) vs. inbound (from the E- to N- zone) navigation within the F-zone. There was no significant difference in decoding accuracy between outbound vs. inbound trips (paired *t*-test; $t(39)$ = 1.52, p=0.136), indicating that the stability of spatial encoding was maintained regardless of the moving direction or perceived context (*Figure 2E*).

One potential concern is that the decoder might rely on visual cues or zone-specific behaviors (e.g. Lobsterbot confrontation or licking) rather than spatial signal per se. To rule out this possibility, we analyzed a subset of data collected when the compartment door was closed, preventing visual access to the Lobsterbot and sucrose port and limiting active foraging behavior. Even under these conditions, the regressor decoded distance with a MAE of 12.14 (±3.046) cm (paired *t*-test;

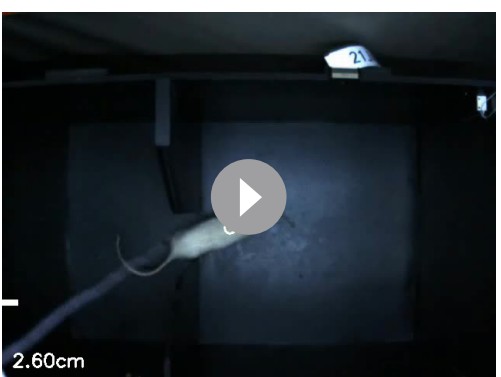

**Video 3.** Example with spatial decoding overlay. The animal's tracked location (tail of the arrow) and decoding error (length and direction of the arrow) are overlaid on a bird's-eye view video. Periods when the animal was distracted (not attending to the task) are also labeled.

https://elifesciences.org/articles/93994/figures#video3

*t*(39) = 12.17, p<0.001). Notably, the regressor's performance was significantly higher with this subset than with the full dataset (paired *t*-test; *t*(39) = 9.895, p<0.001). In summary, these findings indicate ensemble activity in the mPFC contains sufficient spatial information, detectable through ML-based decoding.

## Removing the Lobsterbot decreases spatial decoding accuracy

Earlier studies have suggested that PL neurons do not exhibit any spatially correlated firing patterns in an empty round apparatus (*Poucet, 1997*). This absence of spatial correlates may result from a lack of complex goal-oriented navigation behavior, which requires deliberate planning to acquire more rewards and avoid potential threats. To evaluate how the Lobsterbot's presence impacted spatial decoding accuracy, additional single-unit recording experiments (Ctrl-Exp) were conducted and compared with the previous results (Lob-Exp). In this experiment, three rats followed the same protocol until the microdrive implant surgery. After the surgery, unlike the Lob-Exp group, the Ctrl-Exp group returned to the shuttling phase, during which the Lobsterbot was removed. With this protocol, both groups experienced sessions with the Lobsterbot, but the Ctrl-Exp group's task became less complex, as it was reduced to mere reward collection. During a total of nine recording sessions, 112 units were identified and ANN regressors were built and trained using these data. Statistical tests indicated that the original dataset had significantly lower decoding errors (20.48±2.31 cm) compared to the shuffled dataset (22.61±1.95 cm; paired *t*-test; *t*(8) = 6.20, p<0.001). Nevertheless, when the error was compared with that of Lob-Exp, the Ctrl-Exp group showed a higher decoding error (*Figure 2F*; unpaired t-test; *t*(47) = 3.02, p=0.004). This did not result from differences in activity levels, as there was no significant difference in total travel distance between the Lob-Exp and Ctrl-Exp groups (unpaired *t*-test; *t*(47) = 0.07, p=0.941).

Another factor potentially contributing to the increase in decoding errors could be differences in recording quality across experiments. To discriminate the effects, a generalized linear model (GLM) was constructed with predictors of (1) the Lobsterbot's presence, (2) the number of units recorded, and (3) the recording site (PL vs. IL) on decoding error. The results showed that the GLM predicted the outcome variable with significant accuracy (*F*(3,45) = 14.06, p<0.001), explaining 48.38% of the variance in decoding error. The regression coefficients were as follows: presence of the Lobsterbot (2.76, standard error [*SE*]=1.11, *t*=2.49, p=0.016), number of recorded cells (−0.43, SE = 0.08, *t*=5.29, p<0.001), and recording location (0.91, SE = 0.92, p = 0.329). The results indicated that the number of recorded cells significantly influenced decoding accuracy. Most importantly, the presence of the Lobsterbot significantly decreased the overall decoding error, even after accounting for the number of recorded cells. In summary, the removal of the Lobsterbot decreased spatial decoding accuracy.

## Non-navigational behavior reduces the accuracy of decoded location

Although initial analysis showed that the distance of the rat to the center of the E-zone could be decoded with reasonable accuracy, the spatial precision was unevenly distributed throughout the arena (*Figure 3A*). The magnitude of error was highest in the N-zone, followed by the E-zone, and lowest in the F-zone. A one-way repeated measures ANOVA revealed a significant main effect of the zone (*F*(1.68, 65.52)=40.36, p<0.001). Post hoc analysis showed that the MAE in the F-zone (13.86±2.56 cm) was significantly lower than that in the N-zone (22.72±6.38 cm; *t*(39) = 10.69; p<0.001) and the E-zone (19.98±7.30 cm; *t*(39) = 6.34; p<0.001). However, there was no significant difference between the N- and E-zones (*t*(39) = 2.29; p=0.081; *Figure 3B*). To correct for the unequal distribution of location visits (more visits to the F- than to other zones), the regressor was trained using a subset of the original data, which was equalized for the data size per distance range (see Materials and methods). Despite the correction, there was a significant main effect of the zone (*F*(1.16, 45.43)=119.2, p<0.001) and the post hoc results showed that the MAEs in the N-zone (19.52±4.46 cm; *t*(39) = 10.45; p<0.001) and the E-zone (26.13±7.57 cm; *t*(39) = 11.40; p<0.001) had significantly higher errors when compared to the F-zone (14.10±1.64 cm).

In the N-zone, the rats engaged in stereotypic behaviors such as grooming, climbing, and rearing, during which they stopped locomotive behavior (*Figure 3C*). To determine whether these non-navigational behaviors resulted in higher decoding errors, these behaviors inside the N-zone were manually labeled, and the MAEs during navigational vs. non-navigational behaviors were compared. Due to the imbalanced nature of the dataset resulting from varying preference for non-navigational

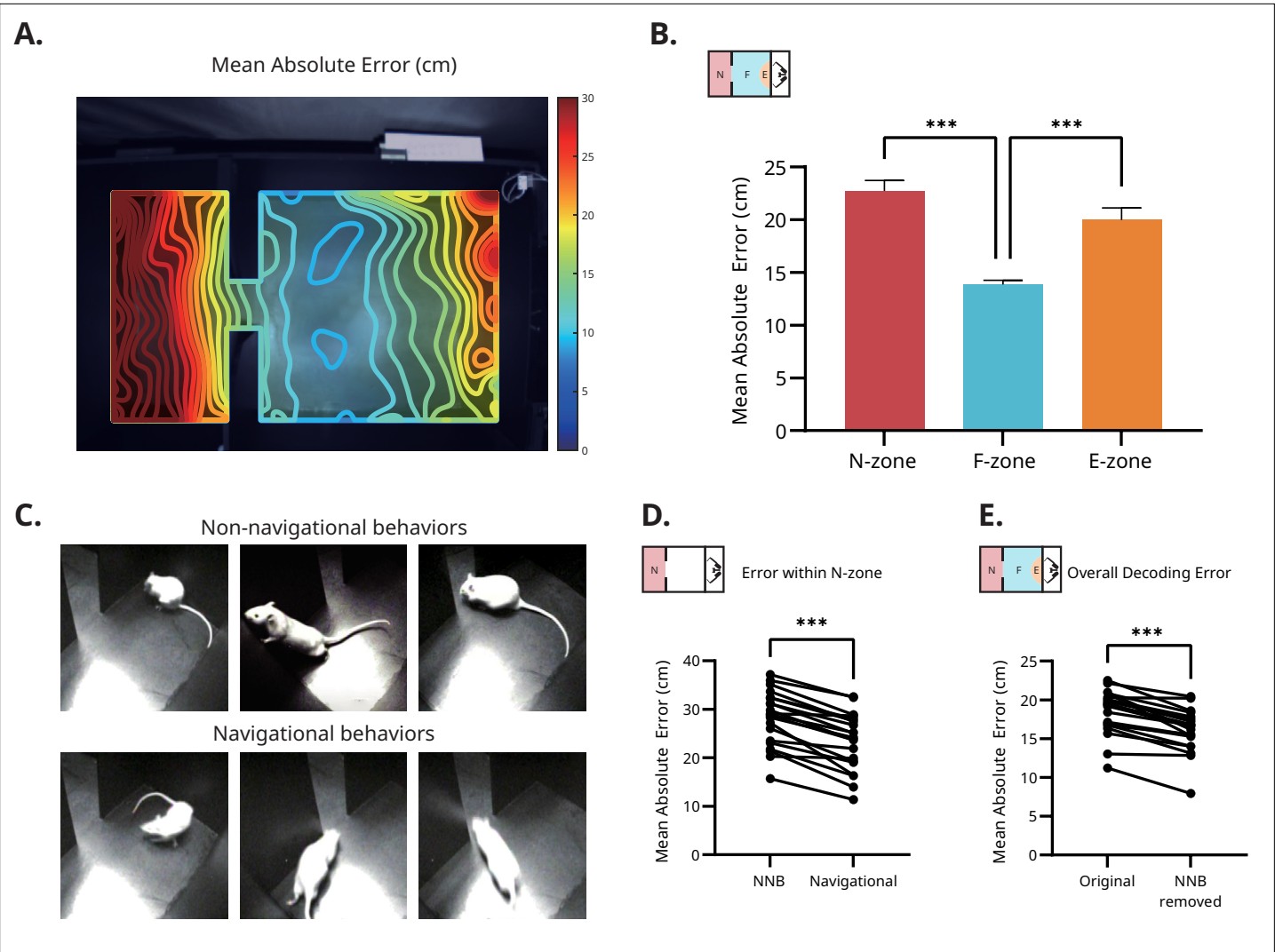

**Figure 3.** Spatial encoding is disrupted by non-navigational behaviors. (**A**) Spatial distribution of the prediction accuracy. The heatmap indicates MAE of a fully trained ANN, superimposed over the entire foraging arena. Prediction accuracy was highest in the F-zone and lower in the N- and E-zone. (**B**) Mean prediction accuracy by the zones. The MAE in the F-zone was significantly lower than the other zones. Error bars represent the SEM. Statistical significance was determined using repeated measure ANOVA with Sidak's multiple comparisions test (***p<0.001). N = 40. (**C**) Examples of the non-navigational behaviors in the N-zone. The top three snapshots depict grooming, rearing, and sniffing. The bottom three snapshots show typical goal-directed navigational movements. (**D**) Comparison of decoding errors (N-zone) during navigational vs. non-navigational behaviors. Errors were significantly larger when the rat was engaged in non-navigational behaviors within the N-zone. Statistical significance was determined using paired *t*-test (***p<0.001). N = 40. (**E**) Comparison between regressors trained with vs. without data from non-navigational behaviors. The overall decoding error was significantly smaller when the regressor was trained without the data from non-navigational behaviors. Statistical significance was determined using paired *t*-test (***p<0.001). N = 40.

behaviors, data from 21 sessions were included, each containing at least 20% of one type of behavior. The paired t-test showed a significant difference between the MAEs during the non-navigational and navigational behaviors in the N-zone (paired t-test; t(20) = 9.381, p<0.001; *Figure 3D*). Moreover, excluding the non-navigational data of the N-zone from the original dataset improved the overall accuracy of the regressor (paired t-test; t(20) = 8.562, p<0.001; *Figure 3E*). These data indicate that erroneous decoding results in the N-zone might have resulted from the disruption induced by non-navigational behaviors.

## Population analysis reveals a different functional mode in the E-zone

To assess the functional coherence of population activity (*Durstewitz et al., 2010*) within and across different zones, principal component analysis (PCA) was performed on the ensemble data collected

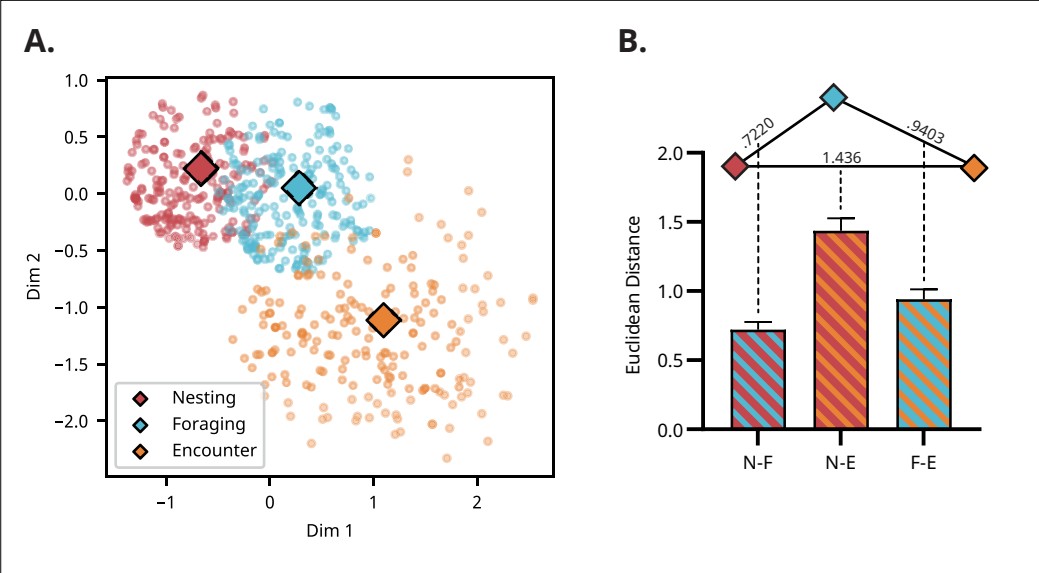

**Figure 4.** PCA results reveal distinctive population activity in the E-zone. (**A**) Representative recording session depicting the first two principal components of population neural activity. Each dot represents a 50 ms segment of multi-unit activity, color-coded by the rat's location. Diamonds mark the centroids of neural representations for each zone. To cluster, only data points near each centroid have been plotted. (**B**) Distances between all centroid pairs across recording sessions. The centroid of the E-zone is distinctly separated from those of the other two zones, indicating a unique neural state within the E-zone. The triangle above the bar plot illustrates the relative distances between the centroids of each zone; longer edges indicate greater dissimilarity between neural ensemble activities. Error bars represent SEM. Statistical significance was determined using Friedman test with Dunn's multiple comparisons test. Pairwise comparisons revealed significant differences between N-F and N-E (p < 0.001) and between N-E and F-E (p < 0.001). N = 40.

The online version of this article includes the following figure supplement(s) for figure 4:

**Figure supplement 1.** Distances between each centroid pairs from all recording sessions.

---

throughout the session. *Figure 4A* illustrates the first two principal components of the transformed neural data from a representative recording session (number of units = 19). First, within- and between-zone distances in population neural activity were compared to determine whether activity within the same zone exhibited similar properties, distinct from those of other zones. A paired *t*-test revealed a significant difference between distances (*t*(119) = 10.46, p<0.001), confirming that the neural activity within a given zone was distinct from those of other zones.

To quantitatively measure functional resemblance among ensemble activity within different zones, we measured the distance between pairs of centroids projected onto the latent space (N-F, N-E, and F-E). A Friedman test showed a significant difference between centroid pairs ($\chi^2(2)$=51.05, p<0.001). Further post hoc analysis revealed a significant difference between N-F and N-E pairs (p<0.001) and N-E and F-E pairs (p<0.001) but not in N-F and F-E pairs (p=0.353), suggesting an ordinal similarity with N-F (D=0.72) being most similar, followed by F-E (D=0.94) and N-E (D=1.44; *Figure 4B*).

Not surprisingly, the firing pattern in the E-zone was the most dissimilar, reflecting a functional state different from those in other zones. In contrast, the distance between centroids of the N- and F-zones was relatively small, indicating that the overlap between the neural subspaces in the two zones was not negligible.

Although PCA can capture population differences between neural ensembles in the zones, a small subset of neurons highly tuned to salient sensory cues could solely drive this separation. Particularly in the E-zone, which encompasses two significant events—head-entry and head-withdrawal—neurons responsive to the delivery of rewards or attacks from the robot could dominate the principal components. To minimize this influence, we defined 'critical event times' as ±1 s from head-entry and head-withdrawal, excluded neural data during these periods, and then recalculated the principal components. The results still demonstrated the same sequential ensemble differences (*F*(1.57, 61.34)=73.56, p<0.001), indicating that the population differences are not driven by events specific

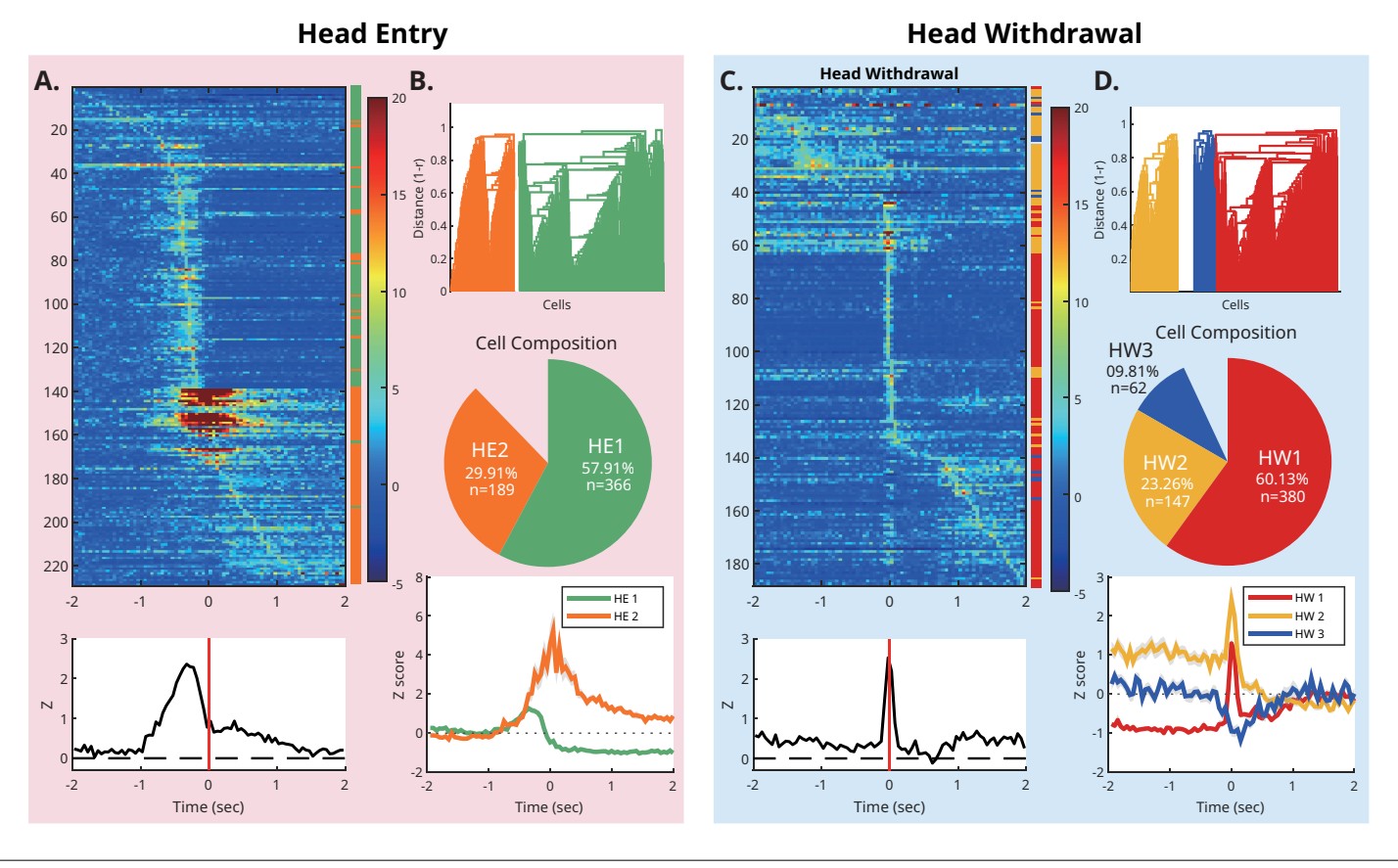

**Figure 5.** Multiple subpopulations in the mPFC react differently to head entry and head withdrawal. (**A**) Top: The PETH of head entry-responsive units is color-coded based on the Z-score of activity. Bottom: The red vertical lines mark the timing of the head-entry. The peak latency of each unit varies from as early as 2 s before to 1~2 s after the head-entry. (**B**) Functional segregation of all recorded units. Top and middle: Two sub-populations of units based on hierarchical cell clustering analysis. Bottom: The averaged activity for each sub-population. (**C**) The PETH of head withdrawal-responsive units is color-coded based on the Z-score of activity. (**D**) Functional segregation of all recorded units. Top and middle: Three sub-populations of units based on hierarchical cell clustering analysis. Bottom: The averaged activity for each sub-population.

The online version of this article includes the following figure supplement(s) for figure 5:

**Figure supplement 1.** A run-and-stop event (sudden velocity drop outside the E-zone) does not evoke neural modulation.

**Figure supplement 2.** Most units are classified into either the HE1-HW1 or HE2-HW2 groups.

**Figure supplement 3.** Type 1 and Type 2 neurons' PETHs around head withdrawal separated by AW and EW.

**Figure supplement 4.** Hierarchical clustering results with different hyperparameter sets.

to any zone (*Figure 4—figure supplement 1*). In summary, populational activity inside the E-zone showed distinctive activity patterns unlike those of active navigation.

## Hierarchical clustering identifies functionally distinctive sub-populations in the E-zone

In the E-zone, navigational behavior was largely replaced by approach attempts and avoidance behaviors. To characterize the temporal dynamics of single-unit activity in the E-zone, we examined the modulation of firing rate to the two key behaviors: head-entry and head-withdrawal. The temporal dynamic of neuronal firing relative to these two behavioral events was visualized using color-coded raster plots and peri-event time histograms (*Figure 5A and C*), featuring spiking activities from all behavior-responsive units (see Materials and methods). Overall, the units showed modulated firing with varying degrees of temporal relations to the two events. Most strikingly, overall modulation patterns were clearly different between head-entry and head-withdrawal.

During head-entry, a large portion of the units increased the firing before the event (*Figure 5A*, top), a pattern confirmed by pooled analysis across neurons (*Figure 5A*, bottom). The mean activity showed a peak approximately 150ms before the head-entry, with the peak exhibiting substantial width (FWHM of ~200 ms), suggesting the presence of varied neuron populations differing in peak position. In contrast, this anticipatory modulation was not clearly identified when the activities were aligned to the head-withdrawal. Instead, activity sharply increased around the event itself (*Figure 5C*, top), producing a narrow peak at 0ms (FWHM = ~75 ms, *Figure 5C*, bottom).

To categorize recorded units into multiple sub-populations based on their functional connectivity in reference to the key events, the unsupervised hierarchical clustering method was applied to all recorded units (*Boehlen et al., 2016*; *Miceli et al., 2020*). We identified two subpopulations when we classified neurons according to their activity around the head-entry (*Figure 5B*). The first sub-population (HE1: n=366, 57.91%) showed an anticipatory increase with a peak located at 350 ms before and a negative modulation after entry and remained low for an extended period. The other sub-population (HE2: n=189, 29.91%) showed a more concentrated positive modulation at around the time of entry. When we used the head withdrawal as a criterion, three sub-populations were identified (*Figure 5D*). The first (HW1: n=380, 60.13%) showed a complex modulation pattern: lower than the baseline in the beginning, followed by a peak firing timed at the head-withdrawal. The second (HW2: n=147, 23.26%) maintained increased activity until the onset of head-withdrawal and returned to zero afterward. The last sub-population, HW3, which consisted of a relatively small number of units (n=62, 9.81%), showed an inhibitory response at the time of head-withdrawal.

Well-trained rats typically run across the F-zone at high velocity and arrive at the sucrose port in the E-zone, stopping abruptly. Moreover, the head-entry is always aligned with the rats' movements, including both subtle (AW) and reflexive (EW) movement. Consequently, it is plausible that the sudden movement might result in a sudden change in neural activity around the time of head-entry and/or head-withdrawal. To exclude this possibility, a 'run-and-stop' event was correlated with the neural data. The event was defined as the transitional moment between a directional locomotion of 2 s or longer and a subsequent pause of 1 s or longer (see Materials and methods). Unit activities were aligned around this run-and-stop event, and the mean activity of each sub-population was plotted for visual inspection. As shown in *Figure 5—figure supplement 1*, there was no detectable neural modulation around the run-and-stop events. The lack of significant modulation at the time of action suggests that the neural modulation occurring around the head-entry and head-withdrawal is not a simple motoric response.

An interesting overlap was found between sub-populations characterized by head-entry and head-withdrawal. In the HW1 group, 78.68% (n=299) of units were classified as HE1. On the other hand, 63.95% (n=94) of the HW2 units were from HE2 (*Figure 5—figure supplement 2A*). Based on this, we further characterized the temporal dynamics of spike activity in relation to the two behavioral events using the combined categories: Type 1 (HE1-HW1) and Type 2 (HE2-HW2). These two subpopulations demonstrated distinctive firing patterns within the E-zone (*Figure 5—figure supplements 2B and 3*). Specifically, Type 1 neurons exhibited a rapid decline in activity following a short peak 200ms before the head-entry, whereas Type 2 neurons displayed a sustained high peak activity ($z \approx 2$) from the onset of the head-entry until the head-withdrawal.

We also examined whether there is a significant difference between the PL and IL in the proportion of Type 1 and Type 2 neurons. In the PL, among 379 recorded units, 143 units (37.73%) were labeled as Type 1, and 75 units (19.79%) were labeled as Type 2. In contrast, in the IL, 156 units (61.66%) and 19 units (7.51%) of 253 recorded units were labeled as Type 1 and Type 2, respectively. A Chi-square analysis revealed that the PL contains a significantly higher proportion of Type 2 neurons ($\chi^2(1, 632)=34.85$, p<0.001), while the IL contains a significantly higher proportion of Type 1 neurons compared to the other region ($\chi^2(1, 632)=18.07$, p<0.001). Taken together, these findings provide support for the existence of at least two distinct subpopulations in the PL and IL that may be involved in different cognitive functions within the E-zone.

## Population activities in the E-zone encode success and failure of avoidance with high accuracy

The previous analysis confirmed that a significant number of neurons modulated their activity before or around the time of approach toward and withdrawal from the E-zone. We then further analyzed the

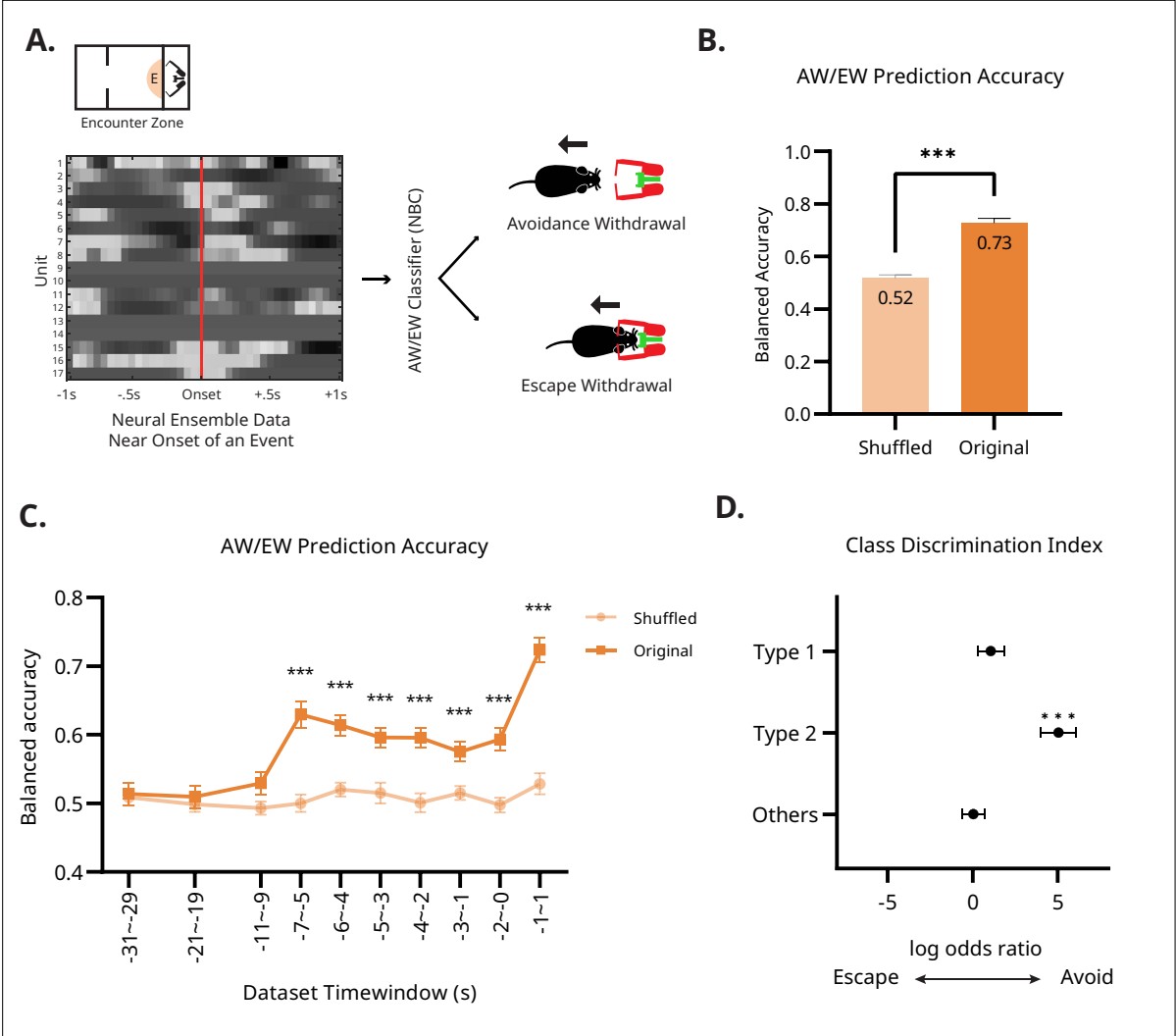

**Figure 6.** Neural ensemble activity predicts failure and success of avoidance response. (**A**) Schematics of the event decoding analysis. The Naive Bayesian decoders are trained with a 2 s window of neural activity to discriminate avoidance or escape on every trial (AW/EW classifier). The grayscale image depicts an example firing pattern of 17 units on a given trial, arranged to the onset of the withdrawal response. The decoder classifies whether this trial is AW or EW based on this data. (**B**) Accuracy of the *naive Bayesian classifier*. The decoding accuracy of the classifier was significantly higher than that from the shuffled data. Statistical significance was determined using paired *t*-test (***p<0.001). N = 40. (**C**) Temporal characteristics of prediction accuracy of the *naive Bayesian classifier*. Prediction accuracy was significantly higher at the time points as early as 5–7 s before the head-withdrawal. Statistical significance was determined using paired *t*-test with Sidak-Bonferroni correction (***p<0.001). N = 40. (**D**) Class discrimination index by the two subpopulations of neurons. The class discrimination index indicates that the Type 2 neurons showed a significant discriminatory power towards AW. Neurons in the Type 2 and the Others group did not exhibit significant discriminatory power. Statistical significance was determined using one sample *t*-test (***p<0.001). N = 299 (Type 1); 94 (Type 2); 239 (Others).

correlation between neural activity and success and failure of avoidance behavior. If the mPFC neurons participate in the avoidance decisions, avoidance withdrawal (AW; withdrawal before the attack) and escape withdrawal (EW; withdrawal after the attack) may be distinguishable from decoded population activity even prior to motor execution. A naive Bayesian classifier (see Materials and methods) was trained to distinguish AW from EW using 2 s neural ensemble data around the onset of the event (*Figure 6A*). We corrected the accuracy against the sample imbalance owing to varying degrees of the AW/EW response ratio across different sessions. Our analysis showed that the classifier based on the original dataset achieved a significantly higher prediction accuracy (72.71%) than a theoretical chance-level prediction (50%) or a shuffled dataset (51.62%; paired *t*-test; *t*(39) = 10.06, p<0.001) in distinguishing AW from EW (*Figure 6B*). Furthermore, we analyzed whether there is a difference in prediction accuracy between sessions with different recorded regions, the PL and the IL. A repeated

two-way ANOVA revealed no significant difference between recorded regions, nor any interaction (regions: $F_{(1, 38)}=0.1828$, p=0.671; interaction: $F_{(1, 38)}=0.1614$, p=0.690).

Because the behavioral reactions following the two events were quite different (bigger reactions during the EW as if the animal failed to predict the attack), it is not clear whether the naïve Bayesian classifier discriminated the two types of responses based on the post-event population activity, which would have been a reflection of the different reactions in the two responses rather than an anticipatory or value-driven encoding. To clarify this issue, we trained naïve Bayesian classifiers using 2 s ensemble activity at varying time points before the execution of the head-withdrawal response. Multiple t-tests between original and shuffled datasets revealed that AW/EW discrimination accuracy began to increase above the chance level 6 s before the actual response and remained significant at all time points (multiple paired *t*-tests with Šídák-Bonferroni correction; ps <0.01; *Figure 6C*, *Supplementary file 1*).

Based on the previous clustering analysis that identified functionally cohesive sub-populations of neurons (Types 1 and 2), we further investigated the predictive encoding by the ensemble activity of the two subpopulations. The naive Bayesian classifier uses class likelihood, P(unit|AW) and P(unit|EW), to classify a particular type of withdrawal response. By comparing the class likelihoods, we can assess the ability of these subgroups to distinguish between AW and EW. This ability is quantified by a 'Class Discrimination Index,' calculated as a log odds ratio. This index reflects the likelihood that a given feature set is associated with a particular type of withdrawal response (AW or EW). To test whether the unit activity right before the head-withdrawal (from 2 s before the head-withdrawal to the onset of the head-withdrawal) was correlated with subsequent avoidance behavior, we calculated the log odds ratio from the trained classifier. The results showed that Type 2 neurons had a significantly higher log odds ratio, indicating a high correlation between unit activity and success in avoidance, whereas no discriminatory modulation was observed for Type 1 neurons or others (one sample t-test, $t(298) = 0.564$; p=0.573, $t(94) = 5.63$; p<0.001, $t(238) = 0.531$; p=0.596, Type 1, Type 2, and others, respectively). These results indicate that Type 2 units carry information that is relevant for classifying trial as AW. To summarize, the populational neural activity of the mPFC in rats within the E-zone distinguished between two types of head-withdrawal behavior, and increased activity of unit subgroups was strongly associated with the execution of AW.

## Overlapping populations of mPFC neurons adaptively encode spatial information and defensive decision

Through a series of analyses, we found that both distance and foraging decision can be decoded from the same mPFC neural ensemble. PCA results support the notion that the mPFC ensemble activity forms a distinctive functional subspace when the rat is engaged in foraging decision facing the robotic predator in the E-zone. Previous studies have also observed different firing patterns from the same mPFC neural ensembles when the rat switches between tasks (*Durstewitz et al., 2010*; *Nigro et al., 2023*). Therefore, it would be interesting to see whether the distance and foraging decision are encoded by two exclusively subsets of neurons or by the whole ensemble.

To quantitatively assess each neuron's contribution to the distance regressor and event classifier, we calculated the feature importance scores for each decoder (*Figure 7A*). This metric measures the increase in error when a neuron's data is shuffled. For example, if a neuron holds substantial data and is heavily relied upon by the decoder, shuffling its signal will significantly increase the error. We analyzed the permutation feature importance scores for each neuron for both distance regression and event classification (see Materials and methods).

First, for the distance regressor, the median permutation feature importance score was 0.73 cm (*Figure 7B*). This suggests that shuffling one unit's neural activity increases the decoding error by less than 1 cm. Moreover, 95% of units exhibited a permutation feature importance score below 1.998 cm, indicating only a negligible increase. To determine whether a subset of neurons with high feature importance scores solely drives distance decoding, we tested whether removing the top 20% of neurons with the highest scores in a given session affects distance decoding. The results showed that although removal significantly increased the decoding error, the regressor trained with the remaining data still achieved a significantly lower error (17.35 cm; Wilcoxon signed rank test, p<0.001). Next, we applied a similar method to the event classifier. This analysis revealed a median permutation feature importance score of 0.01 and a 95th percentile of 0.06. We also tested the effect of removing the top

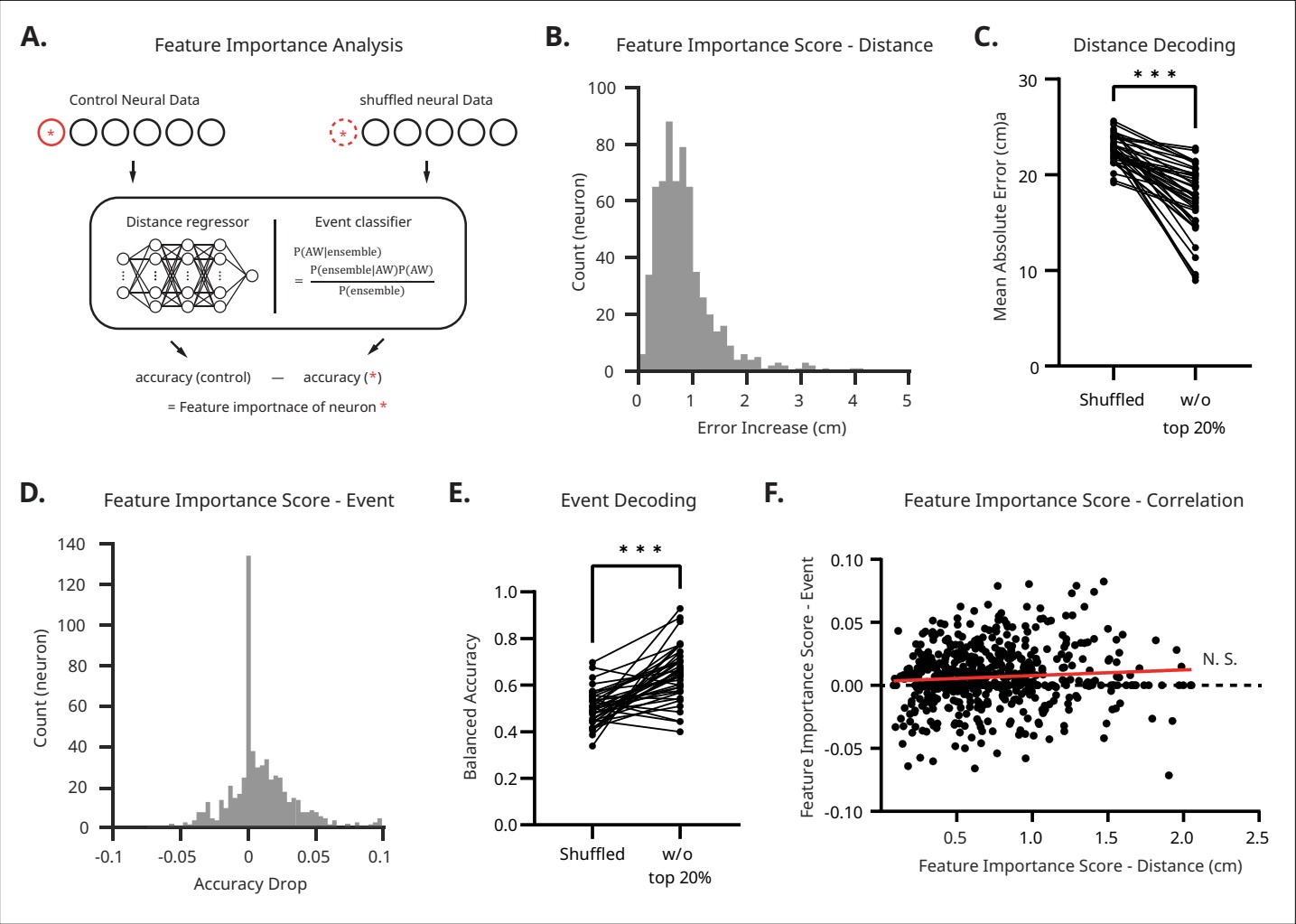

**Figure 7.** Feature importance analysis shows no evidence of dedicated neural subsets for distance or event encoding. (**A**) Schematic of the computational protocol for the feature importance analysis. (**B**) Frequency distribution of all recorded units for their feature importance (measured as error increase) in distance encoding. Only a few units produced a non-negligible increase in error when removed, indicating no dedicated distance-encoding neurons. The red line marks the 95th percentile. (**C**) Accuracy of distance regressor after removing the 20% of high-importance units. Even without these units, the regressor accurately decoded the rat's location. Statistical significance was determined using paired *t*-test (***p<0.001). N = 40. (**D**) Frequency distribution of all recorded units by their feature importance (measured in accuracy drop) in event classification. Only a few units caused a non-negligible accuracy decrease when removed, indicating no dedicated event-encoding neurons. Shuffling unit data resulted in a small decrease in accuracy, and in some cases, an increase. The red line marks the 95th percentile. (**E**) Accuracy of the event classifier after removing the top 20% of high-performance units. Even without these units, the classifier decoded the type of the event (AW vs. EW). Statistical significance was determined using paired *t*-test (***p<0.001). N = 40. (**F**) Correlation between feature importance scores for the distance regressor and the event classifier. No correlation was found between the two measures. N = 632.

20% of neurons with the highest scores and found that the decoder still exhibited a significantly lower error (paired *t*-test, *t*(39)=6.899, p<0.001). This result indicates that there is no sub-population which dominates in encoding distance information.

Furthermore, we tested whether there is a correlation between the feature importance scores of the distance regressor and the event classifier. If there were two subgroups of neurons, each dedicated to encoding distance and events respectively, we might expect to see a negative correlation. To verify this, we calculated Pearson's correlation for each session. We found that there was no significant correlation between the feature importance scores of the two encoders (*r*=0.077, *p*=0.062), indicating the absence of two distinct subpopulations that exclusively encode either distance or event.

Additionally, we found no evidence that Type I or Type II neurons have statistically higher or lower feature importance in the distance regressor. These results support the notion that distance and

avoidance outcomes are not encoded separately by subsets of neurons; rather, the entire ensemble encodes two different concepts according to the given context.

## Discussion

We found distinct ensemble activity states in the mPFC (PL and IL) of a foraging rat, each linked to specific spatial zones or task demands within the foraging environment. The rat's distance from the conflict zone could be decoded with stable accuracy during navigation, but ensemble activity switched to encode defensive behavior when the rat entered the immediate reward-threat zone (E-zone). In this high-conflict context, the rat's threat-evading behavior (AW), coupled with forgoing the reward, could be predicted with a naïve Bayesian classifier. These findings show that mPFC neurons engage in two distinct functional modes of processing allocentric spatial information and egocentric reward-threat information.

In the present study, spatial information was extracted from the mPFC neuron in terms of distance. Previous studies on mPFC spatial coding ranged from reporting no spatial information (*Poucet, 1997*) to precise coordinates encoding with errors as small as a few centimeters (*Ma et al., 2023*; *Mashhoori et al., 2018*). In some studies, the spatial coding mechanism was directly compared between the hippocampus and the mPFC (*Kaefer et al., 2020*; *Powell and Redish, 2014*), and the mPFC cells were found to encode a generalized form of space rather than specific locations. Therefore, we focused on distance as a more generalized spatial index that also reflects an internal representation of value and risk and built an ANN regressor to target it.

Brief navigation was sufficient to recruit most recorded mPFC neurons into spatial encoding, with decoding accuracy consistent across the F-zone regardless of navigational direction (inbound vs. outbound) or visual cues during foraging. The spatial resolution encoded by the ensemble of neurons was precise enough to provide navigational utility (13.86 cm in the F-zone) considering an adult rat's body length (~20 cm without tail), supporting the spatial encoding hypothesis.

However, spatial encoding was disrupted when the rat engaged with the Lobsterbot in the E-zone, paralleling findings that a disfavored reward can distort location decoding (*Mashhoori et al., 2018*). Non-navigational behaviors such as grooming and sniffing also degraded accuracy, suggesting that spatial coding in the mPFC is tightly bound with task demands such as goal-directed navigation or threat avoidance. Consistent with this, removing the Lobsterbot—reducing the task to simple reward collection—lowered decoding accuracy, echoing previous reports that complex, goal-driven navigation sharpens mPFC spatial selectivity (*Hok et al., 2005*; *Ma et al., 2023*).

Consistent with previous findings that mPFC neurons are sensitive to changes in valence—whether reward (*Horst and Laubach, 2013*), threat (*Burgos-Robles et al., 2009*), or both (*Kim et al., 2018*)—the recorded neurons modulated their firing rate around entry into and withdrawal from the E-zone, where sucrose reward and robotic threat coexisted. In line with the mPFC's established role in executive functions (*Dalley et al., 2004*; *Diehl and Redish, 2023*), an ensemble classifier trained on the recorded unit activity predicted AW success with high accuracy (0.73). This prediction remained significantly above the chance level (0.5), even several seconds before the action.

Interestingly, decoding accuracy for distance and avoidance decision did not differ between the PL and IL, despite their well-established functional dissociation in fear conditioning and extinction studies: the PL has been linked to fear expression, and the IL to fear extinction learning (*Burgos-Robles et al., 2009*; *Dejean et al., 2016*; *Kim et al., 2013*; *Quirk et al., 2006*; *Sierra-Mercado et al., 2011*; *Vidal-Gonzalez et al., 2006*). On the other hand, more Type 2 neurons were found in the PL and more Type 1 neurons were found in the IL. To recap, typical Type 1 neurons increased the activity briefly after the head entry and then remained inhibited, while Type 2 neurons showed a burst of activity during head entry and sustained increased activity. One study employing a context-dependent fear discrimination task (*Kim et al., 2013*) also identified two distinct types of PL units: short-latency CS-responsive units, which increased firing during the initial 150ms of tone presentation, and persistently firing units, which maintained firing for up to 30 s. Given the temporal dynamics of Type 2 neurons, it is possible that our unsupervised clustering method may have merged the two types of neurons found in Kim et al.'s study.

We did not observe decreased IL activity during dynamic foraging, whereas prior studies have shown that IL excitability decreases after fear conditioning (*Santini et al., 2008*) and that increased IL activity is required for fear extinction learning. In our paradigm, extinction learning was unlikely,

as the threat persisted throughout the experiment. Future studies with direct manipulation of these subpopulations, particularly examining head withdrawal timing after such interventions, could provide insight into how these subpopulations guide behavior.

We also found that the activity of Type 2 neurons was highly correlated with the success of AW. What is the role of the Type 2 sub-population in the execution of active defensive behavior? One straightforward hypothesis is that the increased activity simply represents internal motivation of fear or anxiety. This premise is supported by multiple experiments where the PL is vital for the expression of freezing responses (*Burgos-Robles et al., 2009*; *Kim et al., 2018*). Consequently, Type 2 units may represent the perceived threat level, which would strongly correlate with the rat's voluntary head-withdrawal. Alternatively, the Type 2 sub-population may contribute to the computation or decision-making processes underlying defensive behaviors. Supporting this, *Halladay and Blair, 2015* identified two distinct sub-populations in the PL selectively responsive to specific defensive behaviors—conditioned movement inhibition and conditioned movement excitation. Given that Type 2 neurons increased firing during AW trials, they may overlap with the excitation-related neuronal subgroups in Halladay and Blair's study. Despite this interesting conjecture, any analysis based on recording data is only correlational, mandating further studies with direct manipulation of the subpopulation to confirm its functional specificity.

Encoded representations in the mPFC range from prediction of upcoming decisions (*Passecker et al., 2019*) to rule switching (*Laskowski et al., 2016*). This diversity of cognitive functions in the mPFC has also been observed in primates, giving rise to the *adaptive coding model* (*Duncan, 2001*). This model posits that the mPFC neurons are fine-tuned to encode task-relevant information depending on the contexts. Our data support this framework: over 80% of units that encoded distance in the F-zone modulated their firing during defensive decision-making in the E-zone, reflecting a context-dependent functional switching. PCA further revealed that the manifold of populational activity in the E-zone was distinct from that of the F- and N-zones, underscoring the heterogeneity of mPFC activity during threat-reward conflict (for review, see *Ebitz and Hayden, 2021*).

Importantly, upon exiting the E-zone, neurons resumed distance encoding in the same manner as before. Because the regressor was trained on data from the entire session, such stable decoding would not be possible if encoding schemes changed each time the rats re-entered the F-zone. In our dynamic foraging task, rats rapidly alternated between navigational behavior in the F-zone and confrontational behavior in the E-zone, yet spatial decoding in the F-zone remained stable. This suggests selective, reversible switching between functional modes rather than global reorganization of coding with each context change.

These results point to a dual, context-sensitive encoding scheme in the mPFC, accommodating both spatial navigation and defensive decision-making within the same behavioral session (*Figure 8A*). This multifunctionality helps reconcile conflicting views of the mPFC's role in spatial orientation by showing that spatial and non-spatial coding can coexist in a task-dependent manner. The 'department store' analogy of the mPFC—where diverse functions emerge from learned associations between contexts, locations, events, and adaptive responses—is consistent with our findings.

In our multi-task design, decoder accuracy varied systematically with context, raising the question of how the mPFC determines which information to encode. We consider three possibilities: (1) a bottom-up hypothesis, in which encoding priorities are dictated by sensory inputs or upstream signals; (2) a top-down hypothesis, in which an internal 'arbitrator' selects the encoding mode and coordinates relevant information; and (3) a competitive hypothesis, in which bottom-up and top-down influences operate simultaneously, with the dominant input determining the active encoding mode (*Figure 8B*).

Although our study cannot resolve the mechanism, a control experiment in which rats performed a simple shuttling task provides some insight. Removing the Lobsterbot degraded spatial encoding rather than enhancing it, suggesting that reducing task demands alone does not bias the mPFC toward one encoding mode. This aligns with the top-down or arbitrator hypothesis, in which mPFC neurons encode heterogeneous information when task demands are high and require coordinated, goal-directed behavior.

Which brain region might act as this arbitrator? Evidence from human neuroimaging studies implicates the anterior cingulate cortex (ACC) as a central hub for switching cognitive modes. During task switching, the ACC shows increased activation (*Hyafil et al., 2009*), enhances connectivity with task-specific regions (*Aben et al., 2020*), correlates with multitask performance (*Kondo et al., 2004*), and

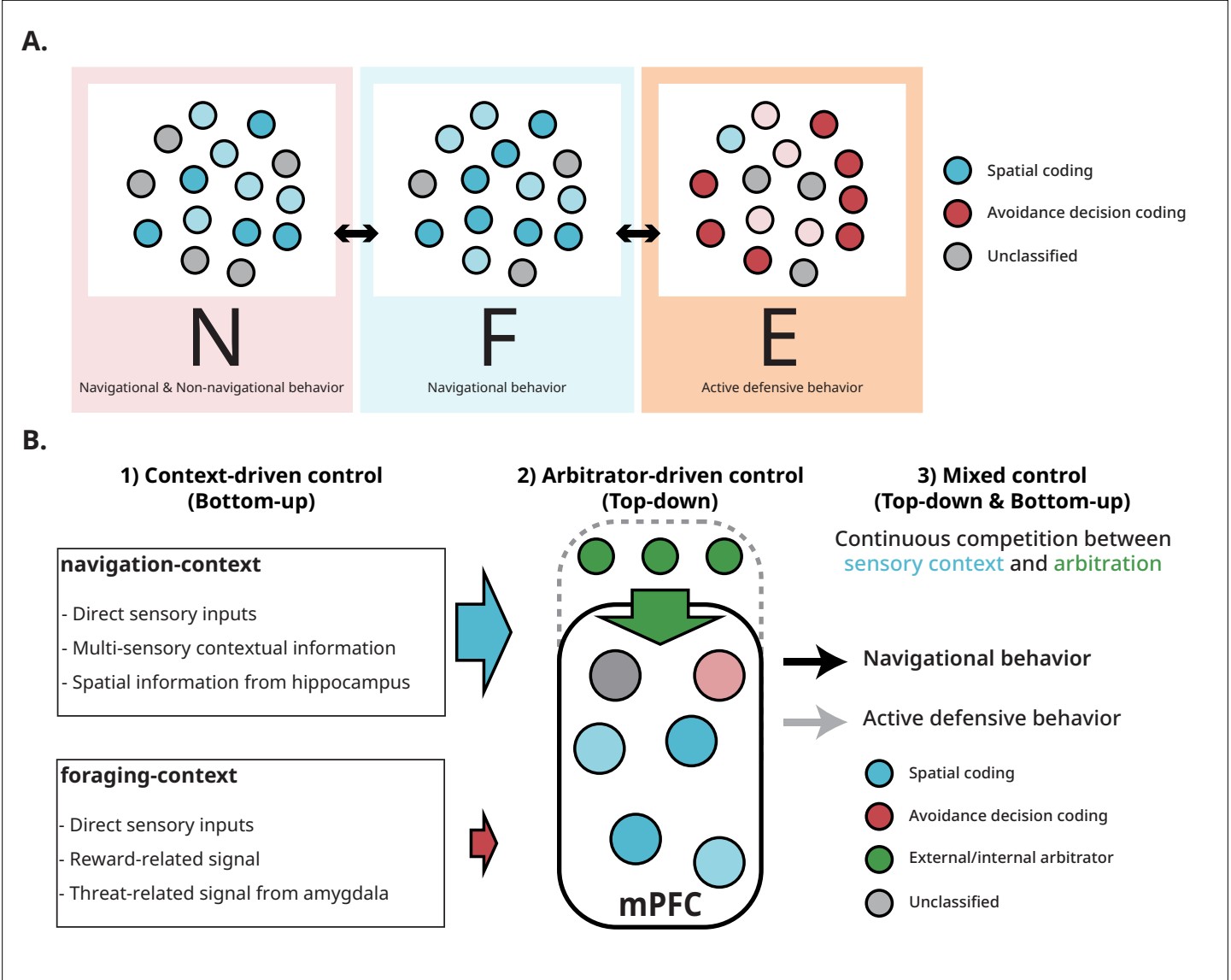

**Figure 8.** Hypothetical control models by which mPFC neurons assume different functional states. (**A**) In the F-zone, navigational behaviors enhance the mPFC's encoding of spatial information compared to other zones. In the N-zone, spatial coding diminishes when the rat engages in non-navigational behaviors. However, in the E-zone, these neurons shift their encoding strategy and become involved in coding for active foraging. We did not find a subset of neurons dedicated exclusively to either spatial coding or active foraging throughout the session. Instead, neurons changed their encoding scheme in a population-wide manner. (**B**) Three hypotheses about how the switch is manifested. In this example, most mPFC neurons encode spatial information (blue circles). Information encoded in the mPFC can be regulated by internal/external arbitration signal (top-bottom blue arrow from green circles), or influenced by direct sensory inputs and navigation-related signals (left-right blue arrow) that prompt mPFC neurons to encode spatial information. A third possibility is that both signals compete to gain control.

monitors the reliability of competing decision systems (*Lee et al., 2014*). Collectively, these findings point to a pivotal role for the ACC in coordinating task assignment. Rodent studies also link the ACC to strategic mode switching (*Tervo et al., 2014*), suggesting that the rodent ACC could similarly arbitrate between strategies, determining which task-relevant variables are represented in the ventral mPFC, including the PL and IL. Future studies combining multi-context tasks with causal manipulations will be essential to determine whether these functional shifts are driven primarily by top-down arbitration or by bottom-up sensory inputs.

## Materials and methods

### Subjects

Eight male Sprague Dawley rats (250–300 g, 5–6 weeks of age, Orient Bio.) were used in this study. The rats were housed in pairs on a 12 hr light/dark cycle with food and water available ad libitum for a 1-week acclimation period. In the following week, rats were handled for 10 min each day until they displayed reduced fear responses towards the experimenter.

### Apparatus and procedures

The apparatus followed the same dimensions as in *Kimm and Choi, 2018*, measuring 800 mm × 480 mm and a height of 500 mm excluding the robot compartment. It was covered in gray EMF-shield fabric to reduce high-frequency noise during experiments. Battery-powered LED strip lighting illuminated the apparatus and −60 dB white noise masked background sounds. The system employed four Arduino Uno boards and a Raspberry Pi board for experiment control and behavioral tracking. All sensor and control signals were relayed to an RV5 multiprocessor (Tucker-Davis Technologies, FL, USA) for further analysis. Bird's-eye view videos were captured during sessions, and the rat's location was tracked using the butter package (https://github.com/knowblesse/butter, copy archived at *Jeong, 2025a*). All control schematics and codes are publicly available (https://github.com/knowblesse/Lobster/tree/master/Apparatus).

Before the training, food and water were gradually restricted until the rats' weight reached 80% of their initial value. Over 3 days, they acclimated to the apparatus for 20 min. Training began with the habituation phase, during which rats had unrestricted sucrose port access for 20 min. The sucrose port has an IR sensor which was activated by a single lick. The rat usually stays in front of the lick port and continuously licks up to a rate of 6.3 times per second to obtain sucrose. Any sucrose droplets dropped in the bottom sink were immediately removed by negative pressure so that the rat's behavior was focused on the licking. Upon reaching 1500 licks per trial, rats proceeded to the 3-day shuttling phase. The shuttling session was 30 min long, and the gate was closed 6 s after the first lick, forcing rats into the N-zone to reopen it. The subsequent Lobsterbot phase introduced the Lobsterbot, which executed two grabbing motions after 3 s (30%) or 6 s (70%) from the initial lick. After three days of Lobsterbot sessions, the rats underwent microdrive implant surgery, and recording data were collected from subsequent sessions, either Lobsterbot or shuttling sessions, depending on the experiment. For all post-surgery sessions, those with fewer than 20 approaches in 30 min were excluded from further analysis.

### Surgery

Before surgery, rats were anesthetized using isoflurane (3–5%) and oxygen (500 mL/min). Enrofloxacin (5 mg/kg) and meloxicam (1 mg/kg) were administered to prevent infection and mitigate postsurgical pain. A craniotomy was performed at AP +3.2 mm and ML 0.5 mm from the bregma using a carbide burr (Komet, Kempten, Germany) mounted on a high-speed drill (Strong 207 A; Saeshin, Daegu, Korea). The tip of each microdrive was lowered to the following coordinates: AP +3.2 mm, ML right 0.3~0.6 mm, DV −4 mm from the bregma. The precise ML coordinate was determined after the craniotomy to avoid damage to the superior sagittal sinus. Six to seven screws were securely affixed to the skull, and two additional screws above the cerebellum were used as the ground and reference points. Finally, dental cement was applied around the microdrive to secure the implant. Rats were allowed at least one week of recovery before the recordings. All procedures were approved by the Korea University Institutional Animal Care & Use Committee (KUIACUC-2019–0036; KUIACUC-2020–0029).

### Data collection

Two types of microdrives were used in the recordings: a custom-made 4-tetrode microdrive and a pre-assembled 16-channel silicon probe (A2x2-tet-10 mm-150-150-121-HZ16_21 mm, NeuroNexus, MI, USA) mounted on a 3D printed microdrive. The microdrive featured an M1.6–18 mm screw with a 0.35 mm pitch. The recording site was retroactively determined by calculating the number of rotations from the final recording site, identified by a marking lesion.

During each recording session, the rat's microdrive was connected to a ZIF-Clip digital headstage (Tucker-Davis Technologies, FL, USA). The neural signals were collected at a 25 kHz sampling rate, amplified, digitized from the head stage, and transferred to the RZ5 (Tucker-Davis Technologies, FL,

USA) for data acquisition. The acquired data stream was further bandpass filtered between 300 and 5000 kHz, and all spikes above 3–4 sigma compared to the stream were collected for further clustering analysis. The Offline Sorter (Plexon, TX, USA) was used for the manual clustering process. All units with firing rates below 0.5 Hz or exactly 60 Hz (power noise) were considered artifacts and removed from further analysis. Sessions with fewer than 10 recorded units were also excluded.

## Spatial decoding data preparation

For the neural data used in the spatial encoding analysis of the mPFC, the following method was employed. First, a Gaussian kernel with a standard deviation of 100ms and a length of 1000ms was prepared. Each unit's spike data were down-sampled and serialized to a 1000 Hz vector, and the kernel was applied to smooth the signal. The mean and standard deviation values of this vector were computed and used to normalize the vector. After normalization, the vector was binned into 50 ms windows, resulting in 20 data points per second. Finally, each datapoint was paired with the corresponding location data.

The spatial firing map was constructed using the same normalized activity data. Neural activity was averaged for each pixel and then smoothed using a Gaussian filter with a sigma of 15 px and a filter size of 1001 px.

## ANN regressor

A neural network regressor was built using the PyTorch package. It comprised seven layers in total, including the output layer; four of these were fully connected (FC) layers and two were drop-out layers. The initial weights for the FC layers were normalized with a mean of zero and a standard deviation of 0.2. Mean squared error loss was used in conjunction with the stochastic gradient descent method. For each regressor, a control regressor was built, which used the same data-label set, but the relationships were randomly shuffled. The evaluation of the regression results used a stratified fivefold cross-validation while ensuring that each subset maintained the same proportion of data from the three zones. The five regression results per dataset were combined, and these results were compared with the true dataset using MAE. For the training/testing dataset, preprocessed 50 ms-long neural data were used. The binned neural data from all units (n) were concatenated to form a 1×n size vector. These vectors were paired with the corresponding distance during the 50ms time window. In the dataset adjusted for uneven location visits, we divided distance values into five equally sized bins. Then, a sub-dataset was created that contains an equal number of data points for each of these bins. Two different GPU servers were employed for training: a Tesla V100 from the National IT Industry Promotion Agency (NIPA, Seoul, Korea) and an in-house A4000.

## Decoding error heatmap construction

To construct the decoding error heatmap, the following method was used. First, error data from each pixel were collected and mean values were computed. To accommodate pixels not visited by a rat during an entire session, an interpolation method was employed. A custom Matlab script, built using the *scatteredInterpolant* function, was used to interpolate errors for unknown locations. The 2D interpolated decoding error was subsequently smoothed using a Gaussian filter with a sigma value of 15 px and a filter size of 1001 px. These smoothed error values were then converted to cm, and a contour plot was overlaid using 25 levels.

## PCA

A custom Python script for PCA was created using the *scikit-learn* package. Neural data vectors generated for the spatial decoding analysis were used for this analysis. The behavior and head tracking data were matched with neural data vectors, and the corresponding zone was labeled to the vectors. All neural data vectors from a single session were projected onto a new space using the PCA. For each zone, the centroid of the projected neural data vectors was calculated using Euclidean distance. The within-zone distance ($d_{within}^{z}\left(x\right)$) and the between-zone distance ($d_{between}^{z}\left(x\right)$) were calculated using the equations below.

$$d_{within}^{z}\left(x\right) = \parallel C^{z} - x^{z} \parallel_{2},$$

$$d_{between}^z (x) = \frac{\sum_{z'=\neg z} \parallel C^{z'} - x^z \parallel_2}{2},$$

where $C^z$ denotes the centroid of the zone $z$ and $x^z$ represents a neural data vector from the zone $z$.

Representative figures were drawn in a manner to highlight the cluster differences. Among all projected neural vectors designated to a zone, the 2000 vectors with the shortest distance from the centroid were selected. From these vectors, 200 were randomly chosen and the first two dimensions of the projected value were plotted.

## Behavior-responsive unit analysis

Behavior-responsive units were identified using the following method. For each trial, spikes within a 4 s window (2 s before and after the onset of three events: gate opening upon returning to the N-zone, head-entry, and head-withdrawal) were counted and registered in 50 ms-long time bins, resulting in 80 data points per trial. These neural activities were then averaged across all trials, producing a 1×80 vector for each event. The binned neural activity ± 2 s from the entrance to the N-zone was chosen as a baseline, considering that the rat was consistently oriented towards areas other than the robot. The mean and standard deviation of the baseline were used to normalize activity during the head-entry and the head-withdrawal. The overall method created two event vectors, head-entry-vector and head-withdrawal-vector, per unit. A unit was considered behavior-responsive to an event if any value within the 80 data points of the event vector was greater or less than 3 sigma.

## Hierarchical cell clustering analysis

To cluster cells that exhibited similar responses during the head-entry and/or head-withdrawal, an unsupervised hierarchical cell clustering method was employed. For every pair of units, the similarity score using Pearson's r was calculated, and the pair with the highest similarity score was grouped to form a new pseudo-unit that had the average response of the two units. This procedure was repeated until all units were grouped into eight pseudo-units. Among each pseudo-unit, if the number of units belonging to the group was equal to or greater than 50, the pseudo-unit was considered a valid cluster. Otherwise, the pseudo-unit was classified as an artifact.

To compensate for the limited number of neurons recorded per session, the hyperparameter set was chosen to generalize their activity and categorize them into major types, allowing us to focus on neurons that appeared across multiple recording sessions. Although changes in the hyperparameter sets resulted in different numbers of clusters, the major activity types remained consistent (*Figure 5— figure supplement 4*). However, there is a chance that this method may not differentiate smaller subsets of neurons, particularly those with fewer than 50 recorded neurons.

## Run-and-stop event analysis

To examine whether a change in movement velocity was related to neural activity, a 'run-and-stop' event was defined by the following criteria: (1) to qualify as 'run' at a given moment, the rat must be moving at least 7.5 cm/s for at least 2 s or longer; (2) to qualify as the 'stop' at a given moment, the rat must be moving at most 5 cm/s for at least 1 s or longer; and (3) the 'run-and-stop' event was the time point between 'run' and 'stop' outside the E-zone. The temporal relationship between the run-and-stop event and neural activity was visualized using the same method as in the behavior-responsive unit analysis.

## Naive Bayes AW/EW classifier and data preparation

For the neural data used in the head-withdrawal-type (AW/EW) classification analysis, a data preparation method similar to that of the spatial decoding analysis was employed. First, the same Gaussian kernel was generated and applied to the down-sampled and serialized neural data. Mean and standard deviation values of this vector were computed for later normalization. Unlike the 50 ms-length neural data used in spatial regression, 2-s-length neural data was used. For each trial, all spikes within a 2 s window around the onset of the head-withdrawal were collected and serialized into a 1000 Hz vector. The Gaussian kernel was applied and normalized using precalculated values. This vector was then binned into 50ms intervals, resulting in 40 data points per event. Binned vectors from all recorded units (n) were concatenated to form a 1 × (40 n) size vector per trial. These vectors were then

paired with their corresponding head-withdrawal-type, AW or EW. In the analysis using varying time windows, the previously employed data preparation method was the same, but the time range of the neural data varied relative to the head-withdrawal.

A naive Bayesian classifier based on the multivariate Bernoulli model was built using the *scikit-learn* package. A custom Python script was written to train and evaluate the classifiers. The classifier employed a uniform prior and smoothing parameter, alpha, which was set to 1 in all classifiers used in the study. Similar to the analysis using the ANN regressor, a control classifier was built that was trained with shuffled data. The evaluation of the classifier results employed the stratified fivefold cross-validation method with each subset maintaining the same proportion of head-entry/head-withdrawal. Five classification results per dataset were combined, and these results were compared with the true dataset using balanced accuracy, calculated using the equation below.

$$\text{balanced accuracy} = \frac{1}{2}\left(\frac{n_{correctly\ labeled\ as\ HE}}{n_{HE}} + \frac{n_{correctly\ labeled\ as\ HW}}{n_{HW}}\right)$$

To interpret how the classifier anticipated upcoming head-withdrawal behavior, class discrimination index was employed, focusing on a dataset with a time window of −2–0 s. To calculate the index, the class likelihood for every dimension was extracted during each test dataset evaluation, and a natural logarithm was applied to balance the small likelihood values. With each unit's neural data divided into 40 bins, the log likelihood values were added to create a joint probability. This process yielded five joint likelihood values per unit, and their average was computed. Following *Mladenić and Grobelnik, 1999* method, the log odds ratio between P(unit|AW) and P(unit|EW) was calculated for all units using the equation below.

$$\text{LogOddsRatio}\left(unit\right) = \log\left(\frac{P\left(unit|AW\right)\left(1 - P\left(unit|EW\right)\right)}{\left(1 - P\left(unit|AW\right)\right)P\left(unit|EW\right)}\right),$$

where P(unit|AW) denotes joint likelihood (probability of observing unit activity given the AW) and P(unit|EW) denotes the same for the EW. These values were then compared according to the unit's assigned group: Type 1, Type 2, or others. Taking the natural log of the odds ratio brings about a symmetricity in the results. A positive value represented a unit's relatively increased activity during the AW compared to the EW, and a negative value represented the opposite.

## Statistical analysis

Statistical analyses were performed using Prism (GraphPad, MA, USA). To confirm normality of the data, the Shapiro-Wilk test was applied to all data we obtained from the experiment. For all statistical tests, p-values of <0.05 were used to assess the significance of differences, and all *t*-tests were two-tailed. The correlation analysis comparing feature importance scores from two decoders excluded the 1% of outliers before computing Pearson's r. Both ANOVA and mixed-effect model analysis applied the Geisser-Greenhouse correction to account for violations of the sphericity assumption. The corrected degree of freedom is presented alongside F values. For multiple comparison tests, Sidak's correction was used to control the family-wise error rate. For multiple linear regression analysis of the distance decoding error, the following model was incorporated.

$$\begin{aligned}\text{MSE} = \beta_0 \quad &+\beta_1(\text{number of units used in the decoder})\\ &+\beta_2(\text{presence of Lobsterbot})\\ &+\beta_3(\text{recording site: PL, IL})\end{aligned}$$

The model fitting method incorporated the least squares method assuming a Gaussian distribution of residuals. Boolean variables, *presence of Lobsterbot*, and *recording site* used dummy codes (0: Lobster, 1: Control; 0: PL, 1: IL) to represent the state of the experiment. All error bars in graphs represent one standard error of the mean. Statistically significant differences are denoted as * for p<0.05, ** for p<0.01, and *** for p<0.001.

## Code availability

We used MATLAB (MathWorks, R2023a, RRID:SCR_001622) and Python Programming Language (version 3.9, RRID:SCR_008394). Python analyses utilized NumPy (RRID:SCR_008633) and scikit-learn (RRID:SCR_002577); specific package versions were not documented. The scripts used for classification and behavioral analysis are openly accessible at the following repository: https://github.com/knowblesse/Lobster (copy archived at *Jeong, 2025b*).

## Acknowledgements

This work was supported by the National Research Foundation of Korea (NRF-2021M3E5D2A01023887, NRF-2017H1A2A1044665) funded by the Korean Government, MSIT. GPU resource was supported by High-performance computing support project by NIPA, Seoul, Korea.

## Additional information

### Funding

| Funder | Grant reference number | Author |
| --- | --- | --- |
| National Research Foundation of Korea | NRF-2021M3E5D2A01023887 | June-Seek Choi |
| National Research Foundation of Korea | NRF-2017H1A2A1044665 | Ji Hoon Jeong |

The funders had no role in study design, data collection and interpretation, or the decision to submit the work for publication.

### Author contributions

Ji Hoon Jeong, Conceptualization, Resources, Data curation, Software, Formal analysis, Funding acquisition, Validation, Investigation, Visualization, Methodology, Writing – original draft, Project administration, Writing – review and editing; June-Seek Choi, Conceptualization, Supervision, Funding acquisition, Validation, Writing – original draft, Project administration, Writing – review and editing

### Author ORCIDs

Ji Hoon Jeong ⓘ https://orcid.org/0000-0002-1214-0989
June-Seek Choi ⓘ https://orcid.org/0000-0002-4394-2140

### Ethics

All surgical, behavioral, experimental procedures were strictly monitored and approved by the Korea University Institutional Animal Care & Use Committee (KUIACUC-2019-0036; KUIACUC-2020-0029).

Reviewer #1 (Public review): https://doi.org/10.7554/eLife.93994.4.sa1
Reviewer #2 (Public review): https://doi.org/10.7554/eLife.93994.4.sa2
Reviewer #3 (Public review): https://doi.org/10.7554/eLife.93994.4.sa3
Author response https://doi.org/10.7554/eLife.93994.4.sa4

## Additional files

### Supplementary files

Supplementary file 1. AW/EW prediction accuracy as a function of dataset time window. Statistical test results for *Figure 6C*, showing avoidance/escape withdrawal decoding accuracy measured across varying dataset time windows.

MDAR checklist

## Data availability

All data including behavioral, neural, and tracking data are available through Dryad (https://doi.org/10.5061/dryad.v9s4mw78c).

The following dataset was generated:

| Author(s) | Year | Dataset title | Dataset URL | Database and Identifier |
|---|---|---|---|---|
| Jeong J, Choi J | 2025 | Population analyses reveal heterogenous encoding in the medial prefrontal cortex during naturalistic foraging | https://doi.org/10.5061/dryad.v9s4mw78c | Dryad Digital Repository, 10.5061/dryad.v9s4mw78c |

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
