## [Editor Report · eLife Assessment]

This **important** study by Jeong and Choi studied neural activity in the medial prefrontal cortex (mPFC) while rats performed a foraging paradigm in which rats forage for rewards in the absence or presence of a threatening object (Lobsterbot). The authors present interesting observations suggesting that the mPFC population activity switches between distinct functional modes conveying distinct task variables- such as the distance to the reward location and types of threat-avoidance behaviors-depending on the location of the animal. The reviewers thought that the results are overall **convincing**, appreciated the value of studying neural coding in naturalistic settings, and felt that this work offers significant insights into how the mPFC operates during foraging behavior involving reward-threat conflict.

---

## [Referee Report · Reviewer #1 (Public review)]

Summary:

In this study, Jeong and Choi examine neural correlates of behavior during a naturalistic foraging task in which rats must dynamically balance resource acquisition (foraging) with the risk of threat. Rats first learn to forage for sucrose reward from a spout, and when a threat is introduced (an attack-like movement from a "LobsterBot"), they adjust their behavior to continue foraging while balancing exposure to the threat, adopting anticipatory withdraw behaviors to avoid encounter with the LobsterBot. Using electrode recordings targeting the medial prefrontal cortex (mPFC), they identify heterogenous encoding of task variables across prelimbic and infralimbic cortex neurons, including correlates of distance to the reward/threat zone and correlates of both anticipatory and reactionary avoidance behavior. Based on analysis of population responses, they show that prefrontal cortex switches between different regimes of population activity to process spatial information or behavioral responses to threat in a context-dependent manner. Characterization of the heterogenous coding scheme by which frontal cortex represents information in different goal states is an important contribution to our understanding of brain mechanisms underlying flexible behavior in ecological settings.

Strengths:

As many behavioral neuroscience studies employ highly controlled task designs, relatively less is generally known about how the brain organizes navigation and behavioral selection in naturalistic settings, where environment states and goals are more fluid. Here, the authors take advantage of a natural challenge faced by many animals - how to forage for resources in an unpredictable environment - to investigate neural correlates of behavior when goal states are dynamic. They investigate how prefrontal cortex (mPFC) activity is structured to support different functional "modes" (here, between a navigational mode and a threat-sensitive foraging mode) for flexible behavior. Overall, an important strength and real value of this study is the design of the behavioral experiment, which is trial-structured, permitting strong statistical methods for neural data analysis, yet still rich enough for unconstrained, natural behavior structured by the animal's volitional goals. The experiment is also phased to measure behavioral changes as animals first encounter a threat, and then learn to adapt their foraging strategy to its presence. Characterization of this adaptation process is itself quite interesting and sets a foundation for further study of threat learning and risk management in the foraging context. Finally, the characterization of single-neuron and population dynamics in mPFC in this naturalistic setting with fluid goal states is an important contribution to the field. Previous studies have identified neural correlates of spatial and behavioral variables in frontal cortex, but how these representations are structured, or how they are dynamically adjusted when animals shift their goals, has been less clear. The authors synthesize their main conclusions into a conceptual model for how mPFC could encode task variables in a context-dependent manner, and provide a useful framework for thinking about circuit-level mechanisms that may support mode switching.

Weaknesses:

The task design in this study is intentionally stimulus-rich and places minimal constraint on the animal to preserve naturalistic behavior, and this introduces some confounds that place some limits on the interpretability of neural responses. For example, some variables which are the target of neural correlation analysis, such as spatial/proximity coding and coding of threat and threat-related behaviors, are naturally entwined. In their revisions, the authors have included extensive analyses and control conditions to disambiguate these confounds. Within the limits of their task design, this provides compelling evidence that mPFC neurons encode threat, decision, and spatial information in a context-dependent manner. Future experiment designs, which intentionally separate task contexts (e.g. navigation vs. foraging), could serve to further clarify the structure of coding across contexts and/or goal states.

While the study provides an important advance in our understanding of mPFC coding structure under naturalistic conditions, the study still lacks functional manipulations to establish any form of causality. This limitation is acknowledged in the text, and the report is careful not to over interpret suggestions of causal contribution, instead setting a foundation for future investigations.

---

## [Referee Report · Reviewer #2 (Public review)]

Summary:

Jeong & Choi (2023) use a semi-naturalistic paradigm to tackle the question of how the activity of neurons in the mPFC might continuously encode different functions. They offer two possibilities: either there are separate dedicated populations encoding each function, or cells alter their activity dependent on the current goal of the animal. In a threat-avoidance task rats procurred sucrose in an area of a chamber where, after remaining there for some amount of time, a 'Lobsterbot' robot attacked. In order to initiate the next trial rats had to move through the arena to another area before returning to the robot encounter zone. Therefore the task has two key components: threat avoidance and navigating through space. Recordings in the IL and PL of the mPFC revealed encoding that depended on what stage of the task the animal was currently engaged in. When animals were navigating, neuronal ensembles in these regions encoded distance from the threat. However, whilst animals were directly engaged with the threat and simultaneously consuming reward, it was possible to decode from a subset of the population whether animals would evade the threat. Therefore the authors claim that neurons in the mPFC switched between two functional modes: representing allocentric spatial information, and representing egocentric information pertaining to the reward and threat. Finally, the authors propose a conceptual model based on these data whereby this switching of population encoding is driven by either bottom-up sensory information or top-down arbitration.

Strengths:

Whilst these multiple functions of activity in the mPFC have generally been observed in tasks dedicated to the study of a singular function, less work has been done in contexts where animals continuously switch between different modes of behaviour in a more natural way. Being able to assess whether previous findings of mPFC function apply in natural contexts is very valuable to the field, even outside of those interested in the mPFC directly. This also speaks to the novelty of the work; although mixed selectivity encoding of threat assessment and action selection has been demonstrated in some contexts (e.g. Grunfeld & Likhtik, 2018) understanding the way in which encoding changes on-the-fly in a self-paced task is valuable both for verifying whether current understanding holds true and for extending our models of functional coding in the mPFC.

The authors are also generally thoughtful in their analyses and use a variety of approaches to probe the information encoded in the recorded activity. In particular, they use relatively close analysis of behaviour as well as manipulating the task itself by removing the threat to verify their own results. The use of such a rich task also allows them to draw comparisons, e.g. in different zones of the arena or different types of responses to threat, that a more reduced task would not otherwise allow. Additional in-depth analyses in the updated version of the manuscript, particularly the feature importance analysis, as well as complimentary null findings (a lack of cohesive place cell encoding, and no difference in location coding dependent on direction of trajectory) further support the authors' conclusion that populations of cells in the mPFC are switching their functional coding based on task context rather than behaviour per se. Finally, the authors' updated model schematic proposes an intriguing and testable implementation of how this encoding switch may be manifested by looking at differentiable inputs to these populations.

Weaknesses:

The main existing weakness of this study is that its findings are correlational (as the authors highlight in the discussion). Future work might aim to verify and expand the authors' findings - for example, whether the elevated response of Type 2 neurons directly contributes to the decision-making process or just represents fear/anxiety motivation/threat level - through direct physiological manipulation. However, I appreciate the challenges of interpreting data even in the presence of such manipulations and some of the additional analyses of behaviour, for example the stability of animals' inter-lick intervals in the E-zone, go some way towards ruling out alternative behavioural explanations. Yet the most ideal version of this analysis is to use a pose estimation method such as DeepLabCut to more fully measure behavioural changes. This, in combination with direct physiological manipulation, would allow the authors to fully validate that the switching of encoding by this population of neurons in the mPFC has the functional attributes as claimed here.

---

## [Referee Report · Reviewer #3 (Public review)]

Summary:

This study investigates how various behavioral features are represented in the medial prefrontal cortex (mPFC) of rats engaged in a naturalistic foraging task. The authors recorded electrophysiological responses of individual neurons as animals transitioned between navigation, reward consumption, avoidance, and escape behaviors. Employing a range of computational and statistical methods, including artificial neural networks, dimensionality reduction, hierarchical clustering, and Bayesian classifiers, the authors sought to predict from neural activity distinct task variables (such as distance from the reward zone and the success or failure of avoidance behavior). The findings suggest that mPFC neurons alternate between at least two distinct functional modes, namely spatial encoding and threat evaluation, contingent on the specific location.

Strengths:

This study attempt to address an important question: understanding the role of mPFC across multiple dynamic behaviors. The authors highlight the diverse roles attributed to mPFC in previous literature and seek to explain this apparent heterogeneity. They designed an ethologically relevant foraging task that facilitated the examination of complex dynamic behavior, collecting comprehensive behavioral and neural data. The analyses conducted are both sound and rigorous.

Weaknesses:

Because the study still lacks experimental manipulation, the findings remain correlational. The authors have appropriately tempered their claims regarding the functional role of the mPFC in the task. The nature of the switch between functional modes encoding distinct task variables (i.e., distance to reward, and threat-avoidance behavior type) is not established. Moreover, the evidence presented to dissociate movement from these task variables is not fully convincing, particularly without single-session video analysis of movement. Specifically, while the new analyses in Figure 7 are informative, they may not fully account for all potential confounding variables arising from changes in context or behavior.

Comments on revisions:

The authors have addressed my previous recommendations.

---

## [Author Response]

The following is the authors’ response to the original reviews

**Public Reviews:**

**Reviewer #1 (Public review):**
Summary:In this study, Jeong and Choi examine neural correlates of behavior during a naturalistic foraging task in which rats must dynamically balance resource acquisition (foraging) with the risk of threat. Rats first learn to forage for sucrose reward from a spout, and when a threat is introduced (an attack-like movement from a "LobsterBot"), they adjust their behavior to continue foraging while balancing exposure to the threat, adopting anticipatory withdraw behaviors to avoid encounter with the LobsterBot. Using electrode recordings targeting the medial prefrontal cortex (PFC), they identify heterogenous encoding of task variables across prelimbic and infralimbic cortex neurons, including correlates of distance to the reward/threat zone and correlates of both anticipatory and reactionary avoidance behavior. Based on analysis of population responses, they show that prefrontal cortex switches between different regimes of population activity to process spatial information or behavioral responses to threat in a context-dependent manner. Characterization of the heterogenous coding scheme by which frontal cortex represents information in different goal states is an important contribution to our understanding of brain mechanisms underlying flexible behavior in ecological settings.Strengths:As many behavioral neuroscience studies employ highly controlled task designs, relatively less is generally known about how the brain organizes navigation and behavioral selection in naturalistic settings, where environment states and goals are more fluid. Here, the authors take advantage of a natural challenge faced by many animals - how to forage for resources in an unpredictable environment - to investigate neural correlates of behavior when goal states are dynamic. Related to his, they also investigate prefrontal cortex (PFC) activity is structured to support different functional "modes" (here, between a navigational mode and a threat-sensitive foraging mode) for flexible behavior. Overall, an important strength and real value of this study is the design of the behavioral experiment, which is trial-structured, permitting strong statistical methods for neural data analysis, yet still rich enough to encourage natural behavior structured by the animal's volitional goals. The experiment is also phased to measure behavioral changes as animals first encounter a threat, and then learn to adapt their foraging strategy to its presence. Characterization of this adaptation process is itself quite interesting and sets a foundation for further study of threat learning and risk management in the foraging context. Finally, the characterization of single-neuron and population dynamics in PFC in this naturalistic setting with fluid goal states is an important contribution to the field. Previous studies have identified neural correlates of spatial and behavioral variables in frontal cortex, but how these representations are structured, or how they are dynamically adjusted when animals shift their goals, has been less clear. The authors synthesize their main conclusions into a conceptual model for how PFC activity can support mode switching, which can be tested in future studies with other task designed and functional manipulations.Weaknesses:While the task design in this study is intentionally stimulus-rich and places minimal constraint on the animal to preserve naturalistic behavior, this also introduces confounds that limit interpretability of the neural analysis. For example, some variables which are the target of neural correlation analysis, such as spatial/proximity coding and coding of threat and threat-related behaviors, are naturally entwined. To their credit, the authors have included careful analyses and control conditions to disambiguate these variables and significantly improve clarity.The authors also claim that the heterogenous coding of spatial and behavioral variables in PFC is structured in a particular way that depends on the animal's goals or context. As the authors themselves discuss, the different "zones" contain distinct behaviors and stimuli, and since some neurons are modulated by these events (e.g., licking sucrose water, withdrawing from the LobsterBot, etc.), differences in population activity may to some extent reflect behavior/event coding. The authors have included a control analysis, removing timepoints corresponding to salient events, to substantiate the claim that PFC neurons switch between different coding "modes." While this significantly strengthens evidence for their conclusion, this analysis still depends on relatively coarse labeling of only very salient events. Future experiment designs, which intentionally separate task contexts (e.g. navigation vs. foraging), could serve to further clarify the structure of coding across contexts and/or goal states.Finally, while the study includes many careful, in-depth neural and behavioral analyses to support the notion that modal coding of task variables in PFC may play a role in organizing flexible, dynamic behavior, the study still lacks functional manipulations to establish any form of causality. This limitation is acknowledged in the text, and the report is careful not to over interpret suggestions of causal contribution, instead setting a foundation for future investigations.

Thank you for the positive comment. We also acknowledge the inherent drawbacks of studying naturalistic behavior. As you also mentioned in the second round of review, separating navigation and foraging tasks in a larger apparatus, such as the one illustrated below, could better distinguish neural activity patterns associated with these different task types. To address the limitations of the current study, we have revised the report to avoid overinterpretation or unwarranted assumptions, and we appreciate that you have recognized this effort.

**Author response image 1. sa4fig1:** 

**Reviewer #2 (Public review):**
Summary:Jeong & Choi (2023) use a semi-naturalistic paradigm to tackle the question of how the activity of neurons in the mPFC might continuously encode different functions. They offer two possibilities: either there are separate dedicated populations encoding each function, or cells alter their activity dependent on the current goal of the animal. In a threat-avoidance task rats procurred sucrose in an area of a chamber where, after remaining there for some amount of time, a 'Lobsterbot' robot attacked. In order to initiate the next trial rats had to move through the arena to another area before returning to the robot encounter zone. Therefore the task has two key components: threat avoidance and navigating through space. Recordings in the IL and PL of the mPFC revealed encoding that depended on what stage of the task the animal was currently engaged in. When animals were navigating, neuronal ensembles in these regions encoded distance from the threat. However, whilst animals were directly engaged with the threat and simultaneously consuming reward, it was possible to decode from a subset of the population whether animals would evade the threat. Therefore the authors claim that neurons in the mPFC switched between two functional modes: representing allocentric spatial information, and representing egocentric information pertaining to the reward and threat. Finally, the authors propose a conceptual model based on these data whereby this switching of population encoding is driven by either bottom-up sensory information or top-down arbitration.Strengths:Whilst these multiple functions of activity in the mPFC have generally been observed in tasks dedicated to the study of a singular function, less work has been done in contexts where animals continuously switch between different modes of behaviour in a more natural way. Being able to assess whether previous findings of mPFC function apply in natural contexts is very valuable to the field, even outside of those interested in the mPFC directly. This also speaks to the novelty of the work; although mixed selectivity encoding of threat assessment and action selection has been demonstrated in some contexts (e.g. Grunfeld & Likhtik, 2018) understanding the way in which encoding changes on-the-fly in a self-paced task is valuable both for verifying whether current understanding holds true and for extending our models of functional coding in the mPFC.The authors are also generally thoughtful in their analyses and use a variety of approaches to probe the information encoded in the recorded activity. In particular, they use relatively close analysis of behaviour as well as manipulating the task itself by removing the threat to verify their own results. The use of such a rich task also allows them to draw comparisons, e.g. in different zones of the arena or different types of responses to threat, that a more reduced task would not otherwise allow. Additional in-depth analyses in the updated version of the manuscript, particularly the feature importance analysis, as well as complimentary null findings (a lack of cohesive place cell encoding, and no difference in location coding dependent on direction of trajectory) further support the authors' conclusion that populations of cells in the mPFC are switching their functional coding based on task context rather than behaviour per se. Finally, the authors' updated model schematic proposes an intriguing and testable implementation of how this encoding switch may be manifested by looking at differentiable inputs to these populations.Weaknesses:The main existing weakness of this study is that its findings are correlational (as the authors highlight in the discussion). Future work might aim to verify and expand the authors' findings - for example, whether the elevated response of Type 2 neurons directly contributes to the decision-making process or just represents fear/anxiety motivation/threat level - through direct physiological manipulation. However, I appreciate the challenges of interpreting data even in the presence of such manipulations and some of the additional analyses of behaviour, for example the stability of animals' inter-lick intervals in the E-zone, go some way towards ruling out alternative behavioural explanations. Yet the most ideal version of this analysis is to use a pose estimation method such as DeepLabCut to more fully measure behavioural changes. This, in combination with direct physiological manipulation, would allow the authors to fully validate that the switching of encoding by this population of neurons in the mPFC has the functional attributes as claimed here.I wanted to add a minor comment about interpreting the two possible accounts presented in fig. 8 to suggest a third possibility: that both bottom-up sensory and top-down arbitration mechanisms can occur simultaneously to influence whether the activity of the population switches. Indeed, a model where these inputs are balanced or pitted against each other, so to speak, to continuously modulate encoding in the mPFC seems both adaptive and likely. Further, some speculation on the source of the 'arbitrator' in the top-down account would make this model more tractable for future testing of its validity.

We thank the reviewer for highlighting this important perspective. We fully agree that an intricate and recurrent interaction between bottom-up and top-down modulations is a highly plausible account of how the mPFC changes its encoding mode. In line with this suggestion, we have incorporated this idea as a third possibility in the revised Discussion, alongside an updated version of Figure 8 that explicitly illustrates this competitive model.

Although we were unable to identify a definitive study directly measuring how the mPFC switches encoding modes across tasks, we did find relevant human EEG and fMRI studies addressing this issue. Based on these findings, we now propose the anterior cingulate cortex (ACC) as a potential hub for top-down arbitration. We have added a paragraph in the Discussion describing this possibility and its implications for future testing.

“Which brain region might act as this arbitrator? Evidence from human neuroimaging studies implicates the anterior cingulate cortex (ACC) as a central hub for switching cognitive modes. During task switching, the ACC shows increased activation (Hyafil et al., 2009), enhances connectivity with task-specific regions (Aben et al., 2020), correlates with multitask performance (Kondo et al., 2004), and monitors the reliability of competing decision systems (Lee et al., 2014). Collectively, these findings point to a pivotal role for the ACC in coordinating task assignment. Rodent studies also link the ACC to strategic mode switching (Tervo et al., 2014), suggesting that the rodent ACC could similarly arbitrate between strategies, determining which task-relevant variables are represented in the ventral mPFC, including the PL and IL. Future studies combining multi-context tasks with causal manipulations will be essential to determine whether these functional shifts are driven primarily by top-down arbitration or by bottom-up sensory inputs.”

**Reviewer #3 (Public review):**
Summary:This study investigates how various behavioral features are represented in the medial prefrontal cortex (mPFC) of rats engaged in a naturalistic foraging task. The authors recorded electrophysiological responses of individual neurons as animals transitioned between navigation, reward consumption, avoidance, and escape behaviors. Employing a range of computational and statistical methods, including artificial neural networks, dimensionality reduction, hierarchical clustering, and Bayesian classifiers, the authors sought to predict from neural activity distinct task variables (such as distance from the reward zone and the success or failure of avoidance behavior). The findings suggest that mPFC neurons alternate between at least two distinct functional modes, namely spatial encoding and threat evaluation, contingent on the specific location.Strengths:This study attempt to address an important question: understanding the role of mPFC across multiple dynamic behaviors. The authors highlight the diverse roles attributed to mPFC in previous literature and seek to explain this apparent heterogeneity. They designed an ethologically relevant foraging task that facilitated the examination of complex dynamic behavior, collecting comprehensive behavioral and neural data. The analyses conducted are both sound and rigorous.Weaknesses:Because the study still lacks experimental manipulation, the findings remain correlational. The authors have appropriately tempered their claims regarding the functional role of the mPFC in the task. The nature of the switch between functional modes encoding distinct task variables (i.e., distance to reward, and threat-avoidance behavior type) is not established. Moreover, the evidence presented to dissociate movement from these task variables is not fully convincing, particularly without single-session video analysis of movement. Specifically, while the new analyses in Figure 7 are informative, they may not fully account for all potential confounding variables arising from changes in context or behavior.Regarding the claim of highly stereotyped behavior, there are some inconsistencies. While the authors assert this, Figure 1F shows inter-animal variability, and the PETHs, representing averaged activity, may not fully capture the variability of the behavior across sessions and animals. To strengthen this aspect, a more detailed analysis that examines the relationship between behavior and neural activity on a trial-by-trial basis, or at minimum, per session, could help.

We thank the reviewer for this thoughtful recommendation and the opportunity to clarify our use of the term “stereotyped behavior.” By this, we were specifically referring to the animals’ consistent licking behavior in the E-zone, rather than to the latency of head withdrawal, which indeed varied across trials and animals. Because licking tempo and body posture during sucrose consumption were highly consistent, the decision to avoid or stay (AW vs. EW) could not be predicted from overt behavior alone. This consistency strengthens our conclusion that the significant predictive power of the Bayesian decoding analysis reflects intrinsic firing patterns of the mPFC neural network, rather than simple behavioral correlates of avoidance.

We also note that the Bayesian model was conducted on a trial-by-trial basis, and the reported prediction accuracy of 73% represents the average across all individual trials (Figure 6B, C). Thus, the analysis inherently captures variability across trials and animals, directly addressing the reviewer’s concern.

The reviewer is correct that the PETHs shown in Figure 5 are based on session-averaged activity aligned to head-entry and head-withdrawal events. The purpose of this analysis was to illustrate that certain modulation patterns could be grouped into 2–3 distinct categories. While averaged activity can provide insight into collective responses to external events, we agree that trial-based analyses provide a more rigorous demonstration of the link between neural ensemble activity and behavioral decisions. This is precisely why we complemented the PETH analysis with Bayesian decoding, which provides stronger evidence that mPFC ensemble activity is predictive of the animal’s choice to avoid or stay.

Similarly, the claim regarding the limited scope of extraneous behavior (beyond licking) requires further substantiation. It would be more convincing to quantify potential variations in licking vigor and to provide evidence for the absence of significant postural changes.

To address this concern, we quantified licking vigor using the inter-lick interval (ILI) as an indirect index. A lick was defined as the period from tongue contact with the IR beam (Lick-On) to withdrawal (Lick-Off), and the ILI was calculated as the time between a Lick-Off and the subsequent Lick-On. Across all animals, ILIs were clustered within a narrow range with a median of 0.155 s (see Author response image 4, left panel).

We analyzed licking vigor at two levels: within trials and within sessions. Because reduced vigor or satiation would lengthen ILIs, comparing the first half and the last half of ILIs within a trial or within a session provides a sensitive proxy for licking consistency.

Within trials: For each of 2,820 trials, we compared the mean ILI of the first half of licks to that of the second half. The average difference was only ~ 17 ms (middle panel). Across sessions: Trial-averaged ILIs were compared between the first and last halves of each session, yielding a mean difference ~ 1.7 ms per session (right panel).

These analyses demonstrate that rats maintained stable licking vigor whenever they entered the E-zone, regardless of avoidance outcome.

**Author response image 2. sa4fig2:** 

Concerning the ANN model, while I understand the choice of a 4-layer network for its performance, the study could have benefited from exploring simpler models. A model where weight corresponds directly to individual neurons could improve interpretability and facilitate the investigation of dynamic changes in neuronal 'modes' (i.e., weight adjustments) over time.

We fully agree with the reviewer on the importance of biologically interpretable models. While artificial neural networks (ANNs) share certain similarities with neural computation, they are not intended to capture biological realism. For example, the error correction mechanism used in ANNs, such as backpropagation has no direct counterpart in mammalian neural circuits. Although we considered approaches that would link each computational node more directly to the activity of individual neurons, building such a model would require temporally sensitive, mechanistic frameworks (e.g., leaky integrate-and-fire networks) and an extensive behavioral alignment effort, which is beyond the scope of the current study.

Our use of an ANN was intended solely as an analytical tool to uncover hidden patterns in multi-unit activity that may not be detectable with traditional methods. Among various machine-learning algorithms, we selected a four-layer ANN regressor because it achieved significantly lower decoding errors (Supplementary Figure S3) and showed robustness to hyperparameter variation (Glaser et al., 2020). To acknowledge the limitations of this approach and suggest future directions, we have revised the Results section to explicitly discuss these points.

“Among various machine learning algorithms, we selected a robust tool for decoding underlying patterns in the data, rather than to model the architecture of the mPFC. We implemented a four-layer artificial neural network regressor (ANN; see Materials and Methods for a detailed structure), as the ANN achieves significantly lower decoding errors (Supplementary Figure S3) and has robustness to hyperparameter changes (Glaser et al., 2020).”